

# System-level design studies for large rotors

Daniel S. Zalkind[1], Gavin K. Ananda[2], Mayank Chetan[3], Dana P. Martin[4], Christopher J. Bay[5], Kathryn E. Johnson[4,5], Eric Loth[6], D. Todd Griffith[3], Michael S. Selig[2], and Lucy Y. Pao[1]

[1]Department of Electrical, Computer & Energy Engineering, University of Colorado Boulder, Boulder, CO 80309, USA
[2]Department of Aerospace Engineering, University of Illinois Urbana-Champaign, Champaign, IL, USA
[3]Department of Mechanical Engineering, University of Texas at Dallas, Richardson, TX 75080, USA
[4]Department of Electrical Engineering, Colorado School of Mines, Golden, CO 80401, USA
[5]National Wind Technology Center, National Renewable Energy Laboratory, Golden, CO 80401, USA
[6]Department of Mechanical and Aerospace Engineering, University of Virginia, Charlottesville, VA, USA

**Correspondence:** Daniel S. Zalkind (dan.zalkind@gmail.com)

**Abstract.** We examine the effect of rotor design choices on the power capture and structural loading of each major wind turbine component. A steady-state, harmonic model derived from simulations using the NREL aeroelastic code FAST is developed to reduce computational expense while evaluating design trade-offs for rotors with radii greater than 100 m. Design studies are performed, which focus on blade aerodynamic and structural parameters as well as different hub configurations and nacelle

placements atop the tower. The effects of tower design and closed-loop control are also analyzed. Design loads are calculated according to the IEC design standards and used to calibrate the harmonic model and quantify uncertainty.

Our design studies highlight both industry trends and innovative designs: we progress from a conventional, upwind, 3-bladed rotor, to a rotor with longer, more slender blades that is downwind and 2-bladed. For a 13 MW design, we show that increasing the blade length by 25 m while decreasing the induction factor of the rotor increases annual energy capture by 11 % while

constraining peak blade loads. A downwind, 2-bladed rotor design is analyzed, with a focus on its ability to reduce peak blade loads by 10 % per 5 deg. of cone angle, and also reduce total blade mass. However, when compared to conventional, 3-bladed, upwind designs, the peak main bearing load of the up-scaled, downwind, 2-bladed rotor is increased by 280 %. Optimized teeter configurations and individual pitch control can reduce non-rotating damage equivalent loads by 45 % and 22 %, respectively, compared with fixed-hub designs.

## 1  Introduction

Wind turbines are large, dynamic structures that experience significant structural loading on their component parts. Design choices impact the loading on each of these parts. We present a model for the rapid computation of wind turbine design loads, which we use to quantify the effect of design trade-offs associated with different rotor concepts. The economics of wind energy have enabled larger wind turbine sizes, generator ratings, and blade lengths. Longer blades are economical simply because they capture more power more often. A wind turbine's annual energy production (AEP) is the total amount of energy captured by a

wind turbine during one year. Increasing the power capture is the primary driver of reducing the cost of wind energy (COE)

$$COE = \frac{CapEx + OpEx}{AEP}, \tag{1}$$





where capital expenditures (CapEx) and operational expenditures (OpEx) make up the cost of building and running a wind turbine. Our goal is to minimize the cost of wind energy, enabling the sale of more wind turbines in an effort to meet our climate goals.

Operational expenditures are non-negligible, but make up roughly 15 % of the total cost, according to a study of the average
2015 offshore wind turbine (Mone et al., 2015). Capital expenditures include the wind turbine parts and balance-of-station costs. Balance-of-station costs account for about 55 % of the total cost and include electrical infrastructure, assembly, and substructure costs. Wind turbine parts (tower, nacelle, blades, etc.) comprise about 30 % of the overall cost of an offshore, fixed-bottom wind plant (Mone et al., 2015). The small cost contribution of the wind turbine blades, which is only a fraction of the cost of the wind turbine parts, and the significant effect of wind turbine blades on AEP contribute to the economics that
enable larger and larger blades.

However, longer blades require additional structural reinforcement, which increases the blade weight, resulting in larger loads experienced by other wind turbine components: like the hub, main bearing, yaw bearing, and tower. Various innovations have enabled lower weight blades; these innovations are then used to subsequently design larger blades that capture more power. Still, the wind turbine components must survive extreme structural loading and last 20-30 years. Wind turbine components are
often designed by various engineering teams based on loads from aeroelastic simulations, making wind turbine design a large, distributed design task.

The aerodynamic and structural aspects of wind turbines must be designed and controlled so that the structural loading for a design is feasible. There is a large inter-dependence between these design aspects (aerodynamic, structural, and controls) and on the various wind turbine components, which has led to numerous design optimization studies. These studies focus primarily on
blade aerodynamic and structural design, e.g., in Ning et al. (2014) and Pavese et al. (2017). Some incorporate dynamic control effects, like Tibaldi et al. (2015) and Bortolotti et al. (2016). System engineering tools, like HAWTOpt2 (Døssing, 2011), WISDEM (Dykes et al., 2014), and Cp-Max (Bortolotti et al., 2016), have been developed to handle the large number of design variables, but often compute structural loads using simplified scaling rules, conservative static calculations, or many nonlinear aeroelastic simulations. A full set of design load cases (DLCs), specified by the International Electrotechnical Commission
(2005) (IEC) in design standards, and simplified for research purposes in Natarajan et al. (2016), can include up to 2000 simulations, which can be costly in terms of computational effort, resulting in long design cycle times. Often the results of these simulations do not fully elucidate the root cause of problematic load cases on the affected turbine component. An attempt to distill the DLCs into a reduced basis for design loads in an optimization framework was presented in Pavese et al. (2016).

We describe an alternative load estimation procedure, based on a set of quasi-steady simulations that reflect the main drivers
of wind turbine loads and the effects of design changes on global wind turbine loads. Since both turbulent and steady wind effects contribute to structural loading and the effect of turbulence has been well studied recently, e.g., in Dimitrov et al. (2018) and Robertson et al. (2018), we will focus our effort on how turbine model changes impact the steady loads caused by wind shear and turbine self-weight. We do this by decomposing the turbine loads into their harmonic components, i.e., the load amplitude of the $i^{\text{th}}$-per-revolution ($i$P) load signal. These signals have been used for control (Bottasso et al., 2013), stability





analysis (Bottasso and Cacciola, 2015), and wind field estimation (Bertelè et al., 2017). Here, we use the same signals to develop a model in order to understand the effect that changing the underlying turbine model has on structural loading.

The power and load estimation procedure developed in this study is used to analyze concepts for enabling rotor radii greater than 100 m. Recently, large rotor concepts have been studied in the European projects UpWind and INNWIND. The Danish

Technical University (DTU) 10 MW Reference Wind Turbine (RWT) (Bak et al., 2013) was provided as a design basis for large rotors to test design methods and tools. The DTU 10 MW RWT has motivated studies that focus on optimization methods (Zahle et al., 2015) and active (McWilliam et al., 2018) and passive (Pavese et al., 2017) load control methods, but the resulting designs from these studies to not deviate far from the base rotor model. A two-bladed, downwind, teetering hub configuration of the DTU 10 MW RWT was developed, which shows that a teetering hub can greatly reduce the unbalanced loading on the main

shaft and blade root (Bergami et al., 2014). However, the authors suggest that the tower stiffness distribution needs to be redesigned in order to avoid a resonance at the twice-per-revolution (2P) rotor harmonic and that 2-bladed rotors (without teeter) increase loading on the main shaft significantly.

A couple of 20 MW rotor designs have been proposed in the literature. Sieros et al. (2012) and Peeringa et al. (2011) use classical similarity scaling rules to upscale conventional turbines. Both conclude that loads due to self-weight will increase

significantly with blade length and drive component design as turbines grow larger. Specifically, edgewise blade loads and the effect of wind shear are magnified for larger rotor sizes.

A series of design studies at Sandia National Laboratories (SNL) detailed the structural design of a 100 m blade, with the goal of reducing the blade mass. First, a classically upscaled blade was given a detailed composite layup and tested against DLCs (Griffith and Ashwill, 2011). Next, a series of design innovations reduced the blade mass from 76 metric tons to 49 met-

ric tons, utilizing carbon fiber reinforcement (Griffith, 2013a), advanced core materials (Griffith, 2013b), and flatback airfoils (Griffith and Richards, 2014).

Another concept to reduce mass-scaling issues is a highly coned, downwind rotor, which has shown that blade loads can be reduced by converting large cantilever loads at the blade root into tensile loads along the span of the blade (Ichter et al., 2016; Loth et al., 2017b). We will analyze this concept and its effect on the structural loading of the other wind turbine components

besides the blades.

There are few openly published documents that quantify the effects of significant design changes and detailed rotor upscaling on the various wind turbine components. We will quantify the effect of aerodynamic changes, including the blade length, axial induction, cone angle, and number of blades, as applied to both upwind and downwind rotors. A simplified structural model will demonstrate the effect of structural reinforcement on blade mass and loads. The upscaled structural model must provide

enough stiffness to compensate for the increasing edgewise blade loads of large rotors. We quantify the effect of changes to the hub by looking at 3-bladed and 2-bladed rotor configurations, and consider the relative benefits of a teeter hinge or individual pitch control for the latter. Finally, we show how the nacelle placement atop the tower and control schemes can impact the loads on the tower and yaw bearing.

We believe this study will contribute an early stage design model for evaluating design concepts more quickly in simulation

by eliminating hundreds of DLC simulations. The simplified load model provides a qualitative understanding of the relationship



between wind turbine structural loads as they progress from the blades to the substructure, highlighting the wind speeds where peak and fatigue loads are most problematic. We calibrate the model against a full set of operational design load cases and quantify the uncertainty. Quantitative design studies evaluate the effect of increased blade size and power capture on global wind turbine loads, as well as the design trade-offs associated with 2-bladed wind turbines, teeter hinges, and individual pitch

control.

We will present the baseline models used for comparison and our general design direction in Sect. 2. Section 3 will outline the tools used for design and simulation and will also provide environmental site specifics. A description of the control scheme used throughout the article is presented in Sect. 4. The quasi-steady harmonic model is described in Sect. 5 and in Sect. 6, the model is calibrated and its uncertainty is quantified. The set of design studies is described in Sect. 7, leading to studies

of blade loads and power capture (Sect. 8), hub and main bearing loads (Sect. 9), yaw bearing loads (Sect. 10), and tower loads (Sect. 11). A discussion of the model's limitations and potential use is provided in Sect. 12, followed by conclusions in Sect. 13.

## 2   Baseline models and design direction

It is useful to start from established designs when doing comparative analysis. In Sect. 8.2, in lieu of a full structural layup

design, we will use these baseline models for scaling the distributed structural properties of rotor blades. For 3-bladed rotors, we will use a conventional rotor design (CONR-13) as a starting point. The CONR-13 is the culmination of a series of design studies aimed at designing a lightweight 100 m blade; it utilizes flatback airfoils, carbon fiber reinforcement, and advanced core materials to reduce the blade mass below state-of-the-art scaling trends. The full design is described in Griffith and Richards (2014). The distributed blade structural properties of the CONR-13 will be used for all 3-bladed rotors in this study.

A downwind, 2-bladed rotor was developed with similar structural advances, but with the goal of reducing the total blade mass by at least 25 % compared to the CONR-13 (Griffith, 2017). The blade was designed to enable segmentation, ultralight design, and a morphing rotor; we refer to this design as the SUMR-13A. The initial aerodynamic design is presented in Ananda et al. (2018). We have slightly modified the initial design to have a downwind cone angle of 5 deg. for the purposes of the design studies presented later. The distributed structural parameters of the SUMR-13A blade were used as a basis for scaling all 2-

bladed rotors in this study. A summary of both baseline models is shown in Table 1. Both rotors were structurally validated to check strain limits, panel buckling, flutter, and fatigue.

In the remainder of this paper, we will evaluate designs aimed at

1. increasing the energy capture, and

2. reducing the wind turbine component loads.

To reduce the cost of energy in Eq. (1), it is most important to increase energy capture (AEP). Industry trends suggest a continued increase in blade length, leading to greater loads on all turbine components. Structural loads contribute to component design and capital cost (CapEx), but require detailed design and cost models for each individual part. Instead of a detailed cost

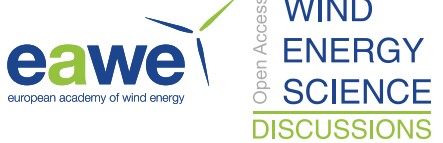

**Table 1.** Turbine models and environmental parameters used throughout this article.

| Turbine Model | CONR-13 | SUMR-13A | SUMR-13B |
|---|---|---|---|
| Rated Power | 13.2 MW | 13.2 MW | 13.2 MW |
| Rated Rotor Speed | 7.44 rpm | 9.90 rpm | 7.99 rpm |
| Rated Wind Speed | 11.3 ms$^{-1}$ | 11.3 ms$^{-1}$ | 10.3 ms$^{-1}$ |
| Hub Height | 142.4 m | 142.4 m | 142.4 m |
| Rotor Radius | 102.5 m | 101.2 m | 125.4 m |
| Rotor Position | Upwind | Downwind | Downwind |
| Blade Mass | 49.5 Mg | 51.8 Mg | 83.2 Mg |
| Number of Blades | 3 | 2 | 2 |
| Max Chord | 5.23 m | 7.22 m | 6.79 m |
| Cone Angle | -2.5 deg. | 5 deg. | 12.5 deg. |

| Environmental Parameters | |
|---|---|
| Wind Turbine Site Class | Class IIB |
| Cut-in, cut-out wind speed | 3, 25 ms$^{-1}$ |
| Mean wind speed at 50 m, hub height | 7.87, 9.11 ms$^{-1}$ |
| Weibull shape, scale factor | 2.17, 10.3 |
| Turbulence Intensity at 15 ms$^{-1}$ | 0.14 |

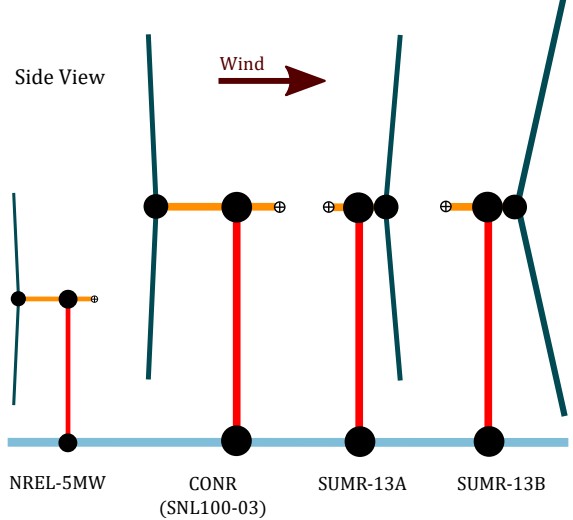

**Figure 1.** Illustrations of the turbines in this study, along with the NREL-5MW reference turbine (Jonkman et al., 2009) for comparison. Tower heights, rotor radii, and cone angles are drawn to scale; overhangs and nacelle center-of-masses are enlarged for comparison.

analysis, which is specific to the component supplier and subject to uncertainty, we will develop a larger rotor design, called the SUMR-13B, described in Sect. 8.1, and then quantify the changes to global wind turbine loads and power capture, while exploring techniques to reduce those loads.

## 3 Design and simulation tools, wind turbine environment

Aerodynamic design was performed using two inverse design tools: PROPID and PROFOIL. PROPID (Selig and Tangler, 1995; Selig, 1995) is an inverse rotor design tool that enables a rotor geometry to be designed based on desired performance specifications like available power, tip speed ratio, wind speed distribution, axial induction, airfoils used, and desired lift distribution along the blade. PROFOIL (Drela and Giles, 1987) is an inverse airfoil design tool. It allows for the design of airfoil geometries based on prescribed velocity distributions and desired geometric (thickness and camber) and aerodynamic properties. Airfoil geometries output using PROFOIL are analyzed using XFOIL (Drela, 1989) and iterated on using PROFOIL until a final converged design is obtained.



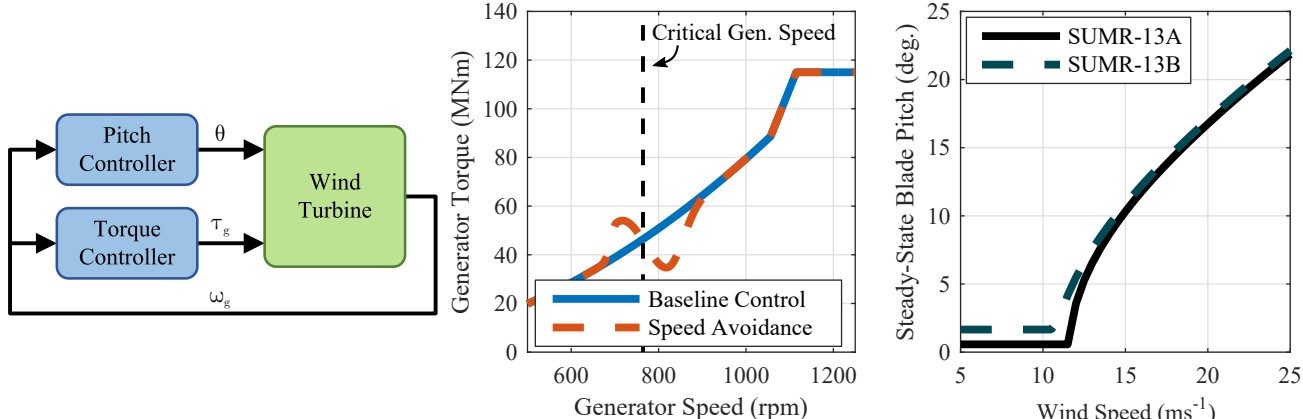

**Figure 2.** Baseline control block diagram, where $\theta$ is the pitch angle, $\tau_g$ is the generator torque, and $\omega_g$ is the measured generator speed (left). The torque control signal (center) for baseline control (blue) and speed avoidance control (red) to avoid the critical generator speed. Steady-state blade pitch angles for the SUMR-13A and SUMR-13B.

    Aeroelastic simulations were performed using the latest version of FAST (Jonkman, 2013). Different FAST modules couple the wind inflow with aerodynamic and elastic solvers that compute the structural loading on the wind turbine. Turbulent wind inputs are generated using TurbSim (Jonkman and Kilcher, 2012). Recent studies have shown that, compared with turbulence, tower shadow effects are relatively small (Noyes et al., 2018) and that tower fairings have been shown to greatly reduce the

tower wake (Larwood and Chow, 2016). Thus, for simplicity, we will assume a tower fairing is present and will omit a tower shadow model from our analysis. Control inputs are provided to FAST through a Matlab/Simulink interface that processes FAST outputs and performs closed-loop control. Fatigue results are computed using MLife (Hayman, 2012), which uses a rainflow counting algorithm to determine load cycles and extrapolates them over the lifetime of the wind turbine.

    To properly compute lifetime fatigue and annual energy production, the wind turbine environment must be provided. The

rotors in this study are all designed to be placed off the coast of Virginia, USA. The site corresponds to a Class IIB turbine rating (International Electrotechnical Commission, 2005), with mean and turbulent wind speed characteristics shown in Table 1.

## 4   Closed-loop control

To simulate turbine design loads and power capture, a closed-loop control scheme is necessary. In below-rated conditions, the generator torque $\tau_g$ is controlled so that the rotor speed $\omega$ is optimal for power capture, following the typical $\tau_g = k\omega^2$ law for

most of the below-rated operating region, before transitioning to above rated (Pao and Johnson, 2011). For simplicity, this is implemented as a look-up table, though more sophisticated methods exist. The look-up table is altered to avoid a critical rotor speed for 2-bladed rotors only (see Fig. 2, center, and Sect. 11 provides more details). In above-rated wind speeds, the pitch angle is controlled to regulate the rotor speed to its rated value using a gain-scheduled proportional-integral (PI) controller. The gains of the PI controller are set so blade fatigue is minimized, subject to a constraint on the maximum generator speed (Zalkind





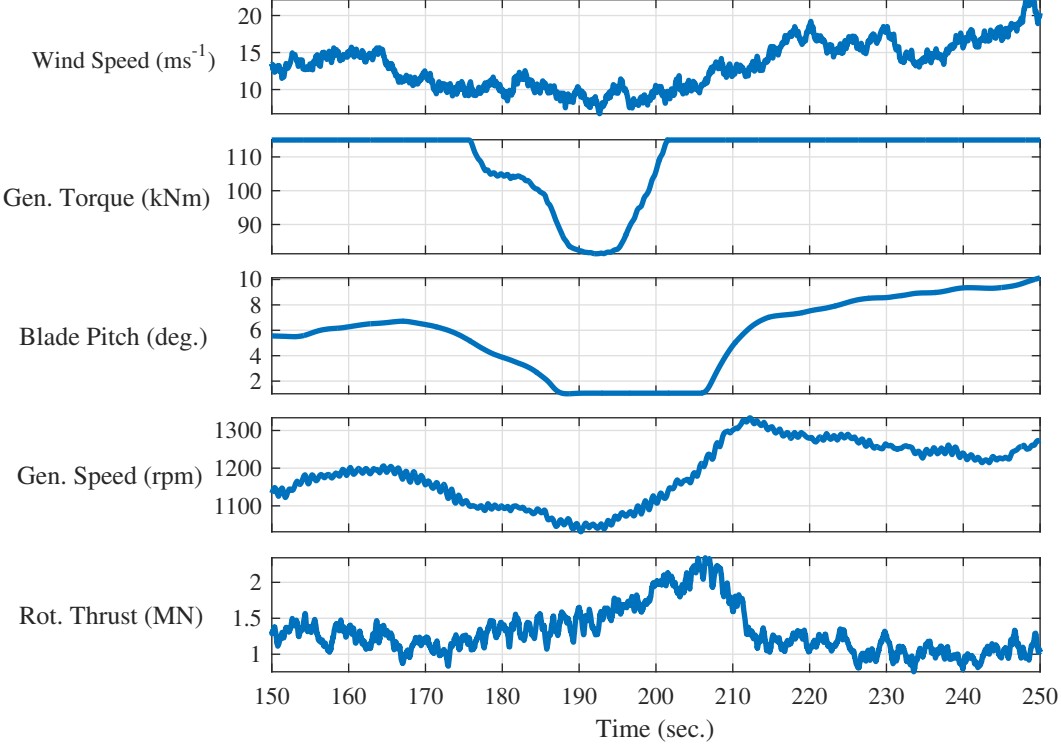

**Figure 3.** Baseline control illustration of a problematic gust for the SUMR-13A baseline rotor in extreme turbulence (DLC 1.3) with a mean wind speed of 14 ms$^{-1}$. The peak rotor thrust near 205 sec. causes the peak blade flapwise load for the SUMR-13A.

et al., 2017). We have chosen this simplified control architecture so that it can be easily tuned for a large number of rotors in the same way. The optimal generator torque control gain $k$ is computed using rotor parameters, and the PI pitch control gains are tuned using a subset of the DLC 1.2 turbulent simulations. The control architecture (as shown in Fig. 2, left) is adapted from the NREL-5MW baseline controller (Jonkman et al., 2009), which is commonly used as a reference to compare new controller designs. While this baseline control may not necessarily be the best possible controller, it allows us to focus on the power and load sensitivity to model changes.

Using closed-loop control for load simulations is important because peak loads often occur near the transition between below- and above-rated operation. With a constant generator rating (13.2 MW), different rotors transition from below to above rated at different wind speeds. Additional control signals, like individual pitch control (IPC) signals, are added to the baseline control signals in Fig. 2 (left).

A controller is also necessary for computing design loads in turbulent DLC simulations, where wind speed changes, or gusts, must be adequately controlled. Often, peak loads are caused by a negative gust, or lull, which we show in Fig. 3. During a decrease in wind speed, the rotor slows and the pitch decreases to its optimal power position. When the decrease in wind speed is followed by a positive gust, the pitch control must react quickly to regulate rotor speed. We model the actuator of





each rotor in this study as a 2$^{\text{nd}}$-order Butterworth filter with a cut-off frequency of 0.25 Hz. This decrease and then increase in wind speed creates a condition where there is an above-rated wind speed, but a below-rated pitch angle setting, resulting in a large thrust force on the rotor and high loads. To capture the effect that closed-loop control has on design loads as rotor changes are made, we use the same control architecture for computing loads in quasi steady-state (Sect. 5) and for turbulent

5   DLC simulations (Sect. 6), updating the controller parameters based on the rotor parameters.

## 5   Harmonic model for load estimation

Load simulations according to the Design Load Cases (DLCs) can be time consuming, so we have developed a simplified model to estimate the loads on wind turbine components more quickly for evaluating design trade-offs across a wide range of parameters. The model runs FAST simulations with a sheared wind inflow such that the wind speed $u$ at height $z$ is

$$u(z) = u_h \left( \frac{z}{z_h} \right)^\alpha, \tag{2}$$

where $z_h$ is the hub height, $u_h$ is the wind speed at hub height, and $\alpha = 0.14$, which is representative of an offshore wind field. Because of the wind shear, the turbine's structural load signals contain harmonic components that depend on the rotor azimuth $\psi$, i.e., a load signal $m(\psi)$ can be expressed as

$$m(\psi) = m_0 + m_c^{1\text{P}} \cos(\psi) + m_s^{1\text{P}} \sin(\psi) + ... + m_c^{i\text{P}} \cos(i\psi) + m_s^{i\text{P}} \sin(i\psi) + ... \tag{3}$$

15   The components are computed by

$$m_0 = \frac{1}{2\pi N_R} \int_{\psi - 2\pi N_R}^{\psi} m(\psi) d\psi, \tag{4}$$

$$m_c^{i\text{P}} = \frac{1}{\pi N_R} \int_{\psi - 2\pi N_R}^{\psi} m(\psi) \cos(i\psi) d\psi, \tag{5}$$

and

$$m_s^{i\text{P}} = \frac{1}{\pi N_R} \int_{\psi - 2\pi N_R}^{\psi} m(\psi) \sin(i\psi) d\psi, \tag{6}$$

where $N_R$ is the number of rotations used in the calculation. We have found that load signals can be reconstructed closely using the first four harmonics; the most energy is usually in either the 1$^{\text{st}}$, 2$^{\text{nd}}$, or 3$^{\text{rd}}$ harmonic depending on the component (see Table 2) and number of blades.

From the components in Eqs. (5) and (6), the magnitude and phase of each harmonic can be computed,

$$|m^{i\text{P}}| = \sqrt{(m_c^{i\text{P}})^2 + (m_s^{i\text{P}})^2}, \tag{7}$$





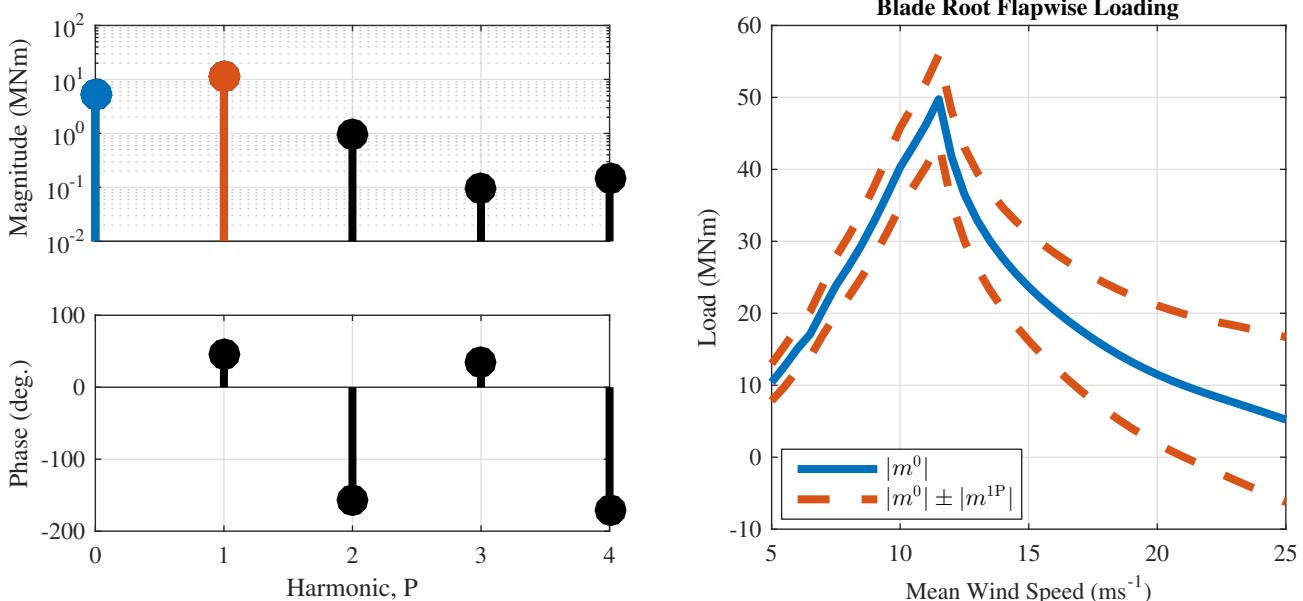

**Figure 4.** Load harmonic magnitude $|m^{i\mathrm{P}}|$ and phase $\phi^{i\mathrm{P}}$ for the zeroth through fourth periodic harmonic of the blade root load in the flapwise direction (left) of the SUMR-13A at 25 ms$^{-1}$. Mean load (blue) superimposed with the 1P harmonic amplitude (red) with respect to wind speed (right) used to estimate fatigue and extreme loads.

and

$$\phi^{i\mathrm{P}} = \tan^{-1}\left(\frac{m_s^{i\mathrm{P}}}{m_c^{i\mathrm{P}}}\right). \tag{8}$$

An example for the blade flapwise load is shown in Fig. 4. We will use these harmonic coefficients, calculated via Eqs. (4)–(8), to estimate fatigue and extreme loads for the various wind turbine components.

## 5.1 Extreme and fatigue loads

The forces and moments on a component drive its design: larger loads require greater reinforcement, leading to greater component mass and cost. We analyze component loads in terms of the maximum (or peak) load

$$m^{\mathrm{Peak}} = \max_{u \in U}(m^0 + m^{n\mathrm{P}}), \tag{9}$$

where $n$ is the dominant harmonic signal component and $U$ is the set of steady-state wind speed simulations in the study. We perform simulations from cut-in to cut-out (Table 1) in 0.5 ms$^{-1}$ increments.

Fatigue loads are computed in terms of the damage equivalent load (DEL): the constant amplitude of a sinusoidal load signal that results in the same total accumulated damage from a more complex load signal. The accumulated damage in simulations with different wind speeds is extrapolated over the turbine lifetime using the wind speed probability distribution

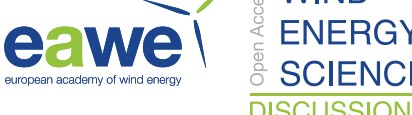



**Table 2.** Structural loads evaluated in this article. Each component has loads in multiple directions and experiences the peak load and greatest contribution to fatigue loads at different wind speeds. $N_B$ denotes the number of blades on the rotor. Loads that are nearly constant across wind speeds do not have a defined peak wind speed (N/A).

| Component | Dominant harmonic | Load direction, name | Wind speed at peak load | Dominant wind speed contributing to fatigue load |
|---|---|---|---|---|
| Blade | 1P | Flapwise, $m_{by}$ | rated | rated |
| | | Edgewise, $m_{bx}$ | N/A | below rated |
| Hub | 1P | Tilt $m_{hy}$ | N/A | rated |
| | | Yaw, $m_{hz}$ | N/A | rated |
| Main Bearing | $N_B$P | Tilt $m_{sy}$ | rated/cut-out | rated |
| | | Yaw, $m_{sz}$ | rated/cut-out | rated |
| Yaw Bearing | $N_B$P | Tilt, $m_{yy}$ | rated/cut-out | rated |
| | | Yaw, $m_{yz}$ | rated/cut-out | rated |
| Tower | $N_B$P | Fore-aft, $m_{ty}$ | rated | tower natural freq. |
| | | Side-to-side, $m_{tx}$ | tower natural freq./cut-out | tower natural freq. |

$p(u)$, characterized by the Weibull distribution in Table 1. We can relate the DEL of a component to its load harmonic by

$$m^{\text{DEL}} = a_{\text{DEL}}(n, w) \sum_{u \in U} p(u) m^{n\text{P}} \tag{10}$$

where $a_{\text{DEL}}$ is a tuning factor that depends on the Whöler exponent $w$ and the dominant harmonic component $n$. The dominant load harmonic $n$P of each component is either 1P or $N_B$P, specified in Table 2, depending on whether the component is rotating (1P) or non-rotating ($N_B$P). Different load harmonics will be specified by their location, direction, and harmonic number, e.g., the 3P main bearing load about the $y_s$-axis will be written $m_{sy}^{3\text{P}}$. The load axes and directions studied in this paper are specified in Table 2 and illustrated in Fig. 5.

## 5.2 Steady versus turbulent loads

The structural loads on a wind turbine originate from both steady-state effects and from dynamics due to turbulence and wind direction changes. In some cases, the effect of turbulence greatly outweighs the steady effects, but in all cases, the steady-state effects predict, to some extent, changes in the design load determined by the DLCs. We quantify this relationship in Sect. 6 by calibrating the steady-state load estimates, computed using Eqs. (9) and (10), to the design loads computed in DLC simulations. In Sects. 7–11, we present the calibrated harmonic load estimates (and their uncertainties) as various turbine design choices are evaluated.



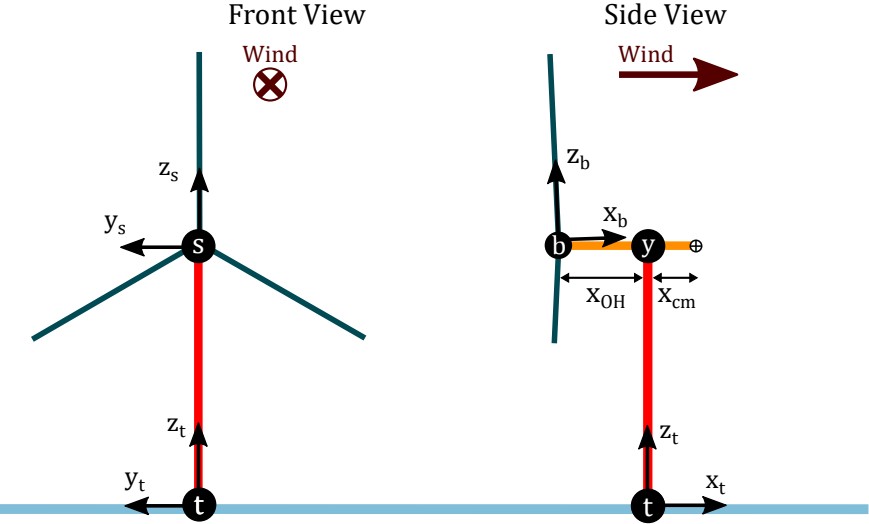

**Figure 5.** Illustration of the load axes used in this article. The non-rotating load axes—tower, main bearing, and yaw bearing—are all parallel and are denoted by subscripts $t$, $s$, and $y$, respectively. Note: the blade, hub, and main bearing axes are collocated; the blade and hub load axes rotate with azimuth angle, as shown in Fig. 12. The CONR-13 is depicted to illustrate the rotor overhang $x_{\mathrm{OH}}$ and nacelle center of mass $x_{\mathrm{cm}}$. The wind is in the same direction as the $x_t$ axis.

## 6  Model calibration and uncertainty

To balance the computational efficiency of the harmonic load estimation in Sect. 5 with the more expensive and realistic design loads computed using DLC simulations, we present the following calibration procedure. In this article, we focus on the power producing design load cases and simulate the following DLCs specified by the IEC standard:

- DLC 1.2: normal turbulence, for fatigue loads, using 6 random seeds at mean wind speeds from cut-in to cut-out, spaced $2\ \mathrm{ms^{-1}}$ apart

- DLC 1.3: extreme turbulence, for peak loads, using the same number of turbulent wind seeds and wind speeds

- DLC 1.4: extreme coherent gust with direction change, for peak loads near rated, above-, and below-rated wind conditions. Different rotor azimuthal initial conditions were simulated to account for the rotor being in different positions when the gust occurs.

- DLC 1.5: extreme wind shear, for peak loads near rated and at cut-out wind speeds. The same azimuthal initial conditions were used as in DLC 1.4.

First, we compare the quasi-steady harmonic load estimates, calculated using the methods in Sect. 5, with the loads computed in DLC simulations. Then, we present a method for calibrating the harmonic load estimates to the design loads. Finally, we





**Figure 6.** Peak main bearing loads computed using DLC simulations versus the quasi-steady harmonic model (top, left) and calibrated load estimates (top, right) for two-bladed rotors (cyan) and three-bladed rotors (magenta). The same color scheme is used to show the relative effect of turbulence on selected component loads (bottom, left), as defined in Eq. (12), and the standard deviation of the error normalized by the mean load is shown for the whole calibration set (bottom, right).

analyze the uncertainty in the calibrated loads, since not all rotors in the design studies of Sects. 7–11 will be simulated using the DLCs. Only a subset of the rotors analyzed in this article, indicated in Table 3, are used in the following procedure to calibrate the harmonic model. The design loads of a free teetering hinge will not be included in the calibration and uncertainty analysis for reasons described in Sect. 9.2; it is marked with an 'x' in Fig. 6.





In Fig. 6 (top, left) we show the design load for the peak main bearing load versus the harmonic load estimate. In general, the harmonic load estimate is much less than the design load computed in DLC simulations. For each component, part of the load can be attributed to the steady loading and part to the turbulent loading:

$$m^{\text{DLC}} = m^{\text{SS}} + m^{\text{turb}}. \tag{11}$$

We quantify the turbulent load contribution of each component load using the turbulence factor

$$f^{\text{turb}} = \frac{\text{mean}(m^{\text{turb}})}{\text{mean}(m^{\text{DLC}})}. \tag{12}$$

Figure 6 (left, bottom) shows the turbulence factor for a selection of the component loads. Some loads, like the edgewise (Blade X) DEL and the hub DEL about the $z_h$-axis for 2-bladed rotors, are more deterministic, with a lower turbulent component, than the others. In general, peak loads are more deterministic than DELs and rotating component loads are more deterministic

than non-rotating component loads. We also see a difference in how turbulence affects 2- vs. 3-bladed rotors, illustrated by the different lines of fit in Fig. 6 (top, left). In general, 2-bladed rotors have a greater turbulent load component, but they also have a larger steady component, so the turbulence factor is similar to 3-bladed rotors. For 3-bladed rotors, the non-rotating load component DELs are not clearly modeled by their quasi-steady harmonic load, so they have a relatively high turbulence factor. Even though some turbine parts have large turbulent components that are not directly modeled in steady state, there is

still good correlation with the steady loads.

We calibrate the harmonic loads estimates by fitting a linear model through the harmonic estimates and design loads

$$m^{DLC} = a_{\text{cal}} m^{SS} + b_{\text{cal}} \tag{13}$$

using a linear least squares estimate of the parameters $a_{\text{cal}}$ and $b_{\text{cal}}$. Because 2- and 3-bladed rotors sample turbulence differently, we define a calibration set $(a_{\text{cal}}, b_{\text{cal}})$ separately for each, illustrated by the different fits of Fig. 6 (top, left). To estimate the

design load, the same calibration set is used:

$$m^{est} = a_{\text{cal}} m^{SS} + b_{\text{cal}}, \tag{14}$$

which results in a calibrated harmonic load estimate equal to the design load, plus some uncertainty (Fig. 6, top, right).

We analyze the uncertainty of the calibrated data set (both the 2- and 3-bladed rotors, without the outlier case) by computing the standard deviation of the error for each component. The standard deviation of error is normalized by the mean load over all

rotors and shown in Fig. 6 (bottom, right). We also indicate the standard deviation of the error for each component load in the figures of Sects. 7–11.

In general, the standard deviation of the error is less than 12 % of the mean value, which indicates decent agreement between the calibrated harmonic load estimates and the DLC-computed design loads. The cases with lowest uncertainty tend to have lower turbulence factors, like the blade edgewise (Blade X) DEL and the hub $z_h$-axis DEL. The AEP is also very well estimated

by the steady-state model, which is good for power capture predictions as long as the effects of turbulence are calibrated.



The most erroneous load component is the peak yaw bearing load, which has a large turbulent component and where a subset of the calibration set (the aerodynamic trade study designs) control a problematic gust event, like the one in Fig. 3, similarly. These rotors have design loads that are about the same for each, despite the differences predicted by the steady-state model. The design loads for this component might be more a function of the gust event than the turbine configuration. The most erroneous value (using the metric of standard deviation normalized by the mean) is the minimum tower clearance, but this is influenced by the small average tower clearance over all rotors. Such an important design parameter would certainly be subject to verification using the full set of DLCs before deeming the tower safe from blade strike.

In the remainder of this article, we use these calibrated harmonic load estimates to analyze the structural loading and power capture of the various rotor configurations in Table 3.

**Table 3.** Set of turbines designed and analyzed in this article. * denotes a turbine for which DLC simulations were performed and used to calibrate the harmonic load estimates to DLC-based design loads. Otherwise, only the harmonic load analysis is performed. [†] was omitted from the calibration set. [‡] denotes the SUMR-13A rotor and [§] denotes a 3-bladed variation of the SUMR-13A rotor.

| |
|---|
| **Baseline Set (Sect. 2):** CONR-13*, SUMR-13A*[‡], SUMR-13B* |
| **Rotor Aerodynamic Trade Studies (2-bladed, Sect. 8.1):** |
| Available rotor power (MW): 13.9*[‡], 14.9, 15.9, 16.9* |
| Axial Induction (-): 0.175*, 0.200, 0.225, 0.250, 0.275, 0.300, 0.333*[‡] |
| Cone Angles (deg.): -5*, 0, 5*[‡], 10, 15, 20* |
| **Rotor Aerodynamic Trade Studies (3-bladed, Sect. 8.1):** |
| Available rotor power (MW): 13.9*[§], 14.9, 15.9, 16.9* |
| Axial Induction (-): 0.175*, 0.200, 0.225, 0.250, 0.275, 0.300, 0.333*[§] |
| Cone Angles (deg.): -5*, 0, 5*[§], 10, 15, 20* |
| **SUMR-13B Structural Parameter Analysis (Sect. 8.2)** |
| $k_{\text{All}} = 0$*, $k_M = 1$, $k_{Fs} = 1$, $k_{Es} = 1$, $k_{\text{All}} = 1$* |
| **SUMR-13B Hub Configurations (Sect. 9)** |
| SUMR-13B (3-bladed)* |
| Teeter: Free[†], Ideal* |
| IPC: Blade*, Bearing* |

## 7   Overview of design studies

In this section, we present the design and simulation results of the 42 turbines shown in Table 3. The design loads for each rotor are estimated and calibrated using the methods in Sect. 5 and Sect. 6, respectively. Additionally, gross annual energy



**Figure 7.** Overview of the design studies performed in this paper. The loads on each component (blue) transfer from the blades to the tower base as shown. Design studies (yellow) that affect each component are performed in Sects. 8–11 by altering the design parameters in green. Rotor design parameters (orange) affect all aspects of turbine design.



production (AEP) is calculated using the generator power $P(u)$ at steady-state wind speed $u$ by

$$AEP = 8760 \sum_{u \in U} p(u)P(u),\qquad(15)$$

where $p(u)$ is the Weibull distribution in Table 1 and 8760 is the number of hours in a year.

We first examine changes to the blade loads and power capture of the SUMR-13A due to variations in the aerodynamics,

including the blade length, axial induction, and cone angles. Both upwind (negative) and downwind (positive) cone angles are evaluated. The aerodynamic changes lead to a larger, heavier, but more powerful SUMR-13B rotor, which we use to study the effect of mass and stiffness scaling on blade loads. Next, non-rotating component loads will be compared for different hub configurations, considering the number of blades, a teetering hinge, individual pitch control, and rotor placement (upwind vs. downwind). Finally, the effect of a downwind rotor on yaw bearing design loads will be presented and the effect of a two-bladed

rotor on tower design will be investigated. A summary of the design parameters considered in this article and the process for incorporating their interconnections is shown in Fig. 7; details are given in Sects. 8–11.

## 8    Blade loads and energy capture

We begin by analyzing the effect of changing rotor aerodynamics on blade loads and energy capture. Blade loads are computed at the blade root in both the flapwise ($m_{by}$) and edgewise ($m_{bx}$) directions. Blade flapwise loads are primarily aerodynamic

in nature and depend on the thrust force exerted on the blades from the wind inflow. Peak blade flapwise loads occur near rated wind speed, which represents the worst combination of wind speed and orthogonal blade surface area, but before the blade begins pitching to regulate power in above-rated operation. Blade pitch has a significant influence on the mean blade flapwise load and control actions can often cause peak loads, e.g., when the pitch angle decreases towards its fine pitch angle to maximize power and then a wind speed gust occurs. The dependence of this load on the control system highlights the necessity

of including control design at an early stage.

Flapwise fatigue loads are driven by blade thrust, wind shear, and, to a small degree, blade weight and cone angle. Edgewise fatigue loads, on the other hand, have a nearly constant load cycle amplitude, unless the rotor speed is rapidly changing. The load cycle amplitude of edgewise blade loads depends on the blade weight, creating a large positive and then negative load when the blade is in each horizontal position during a rotor revolution. Edgewise fatigue loads increase with blade length and

mass and influence the design of the baseline blade structures used in this study (CONR-13, SUMR-13A). Additional stiffness must compensate for increased edgewise loads, but at the cost of increased blade mass, leading to even greater loads. We will explore this relationship in Sect. 8.2.1.

### 8.1    Rotor aerodynamics

We evaluate rotors with longer blade lengths, lower axial induction factors, and large, downwind cone angles, using the SUMR-

13A design described in Sect. 2 as a baseline. These design studies have led us to an updated, larger, 2-bladed design, indicative

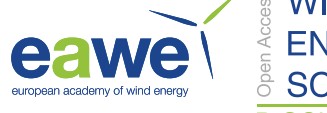

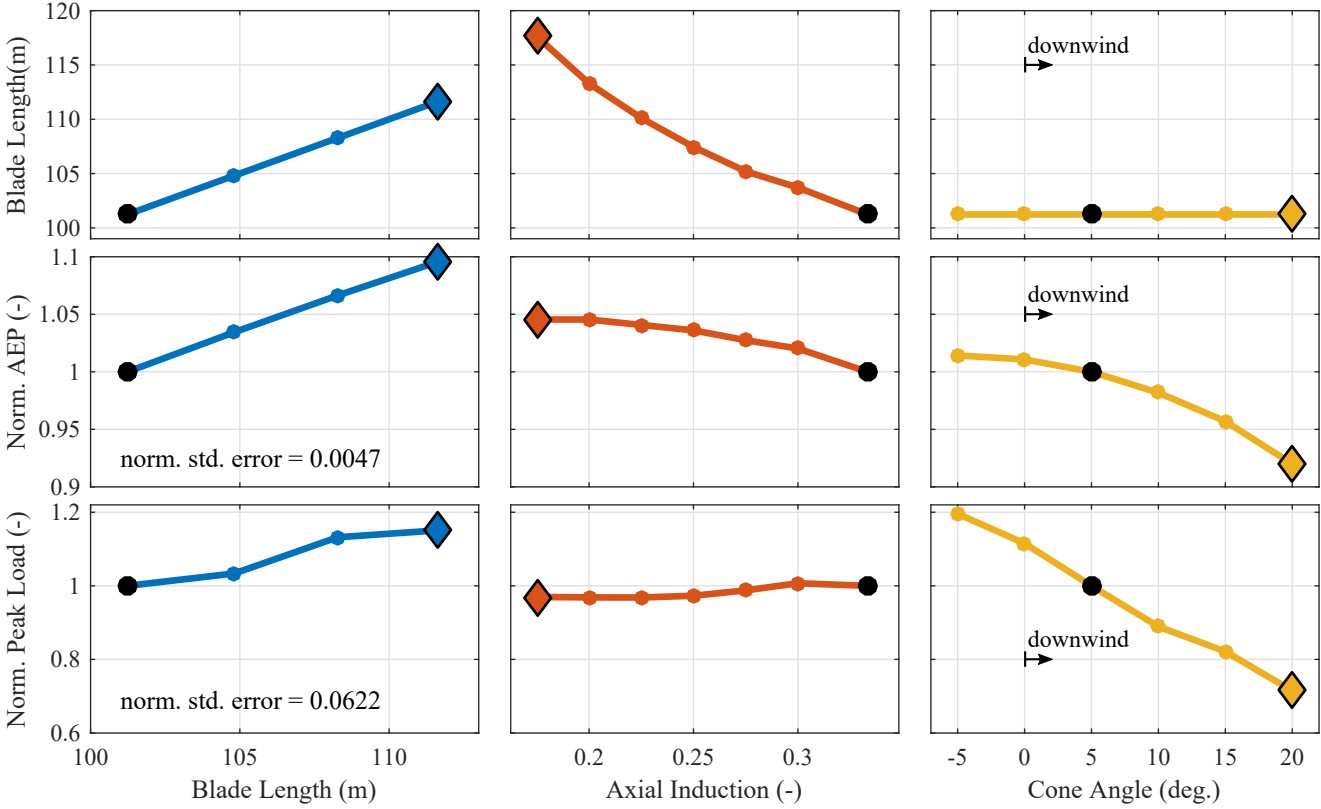

**Figure 8.** Summary of aerodynamic design studies: the blade length, axial induction, and cone angle are varied while the AEP and peak blade load are calculated and compared to the base case (SUMR-13A, black dot in all). The standard deviations of the errors for AEP and peak flapwise load are normalized to the SUMR-13A values and apply across all design studies. All rotors here are 2-bladed and positive cone angles correspond to downwind rotors. Unless otherwise specified, the available rotor power is 13.9 MW, the axial induction is 0.333, and the cone angle is 5 deg.

of the trends in industry towards longer, more slender blades, but with a greater downwind cone angle. We will call this new rotor SUMR-13B (see Table 1 for more details).

Blade length is changed indirectly in PROPID, by increasing the available rotor power at 11.3 ms$^{-1}$ from 13.9 MW to 16.9 MW. However, all rotors are controlled to have the same rated generator power of 13.2 MW, which somewhat constrains
5 peak blade loads by transitioning to above-rated control at lower wind speeds. Since generator power is fixed, there is a linear increase in AEP with blade length (blue, left column in Fig. 8), whereas power at a fixed wind speed increases quadratically with blade length. The increased rotor swept area increases both power capture and blade loads; a 10 % increase in rotor radius results in about a 10 % increase in AEP and 15 % increase in peak blade flapwise load. For the blade length design study, the axial induction factor along the blade is fixed at ⅓ (theoretical Betz limit).





The rotors used to evaluate axial induction (red, center column in Fig. 8) are designed by fixing the flapwise root bending loads to that of the SUMR-13A and fixing the available rotor power at rated wind speed to 13.9 MW. The blade length, chord, and twist are allowed to vary as the local axial induction factor—from the 25 % radial location to the blade tip—varies from 0.175 to 0.3 in increments of 0.025. Decreasing the designed axial induction of the rotor results in longer, more slender blades

that capture more energy while constraining blade loads. In the most extreme example, a blade with a 0.175 axial induction factor can increase the AEP by 5 %, compared to a rotor with aerodynamically optimal blades (axial induction factor of ⅓), but requires 16 % longer blades.

The cone angle design study is performed using the same baseline SUMR-13A blades for each rotor, but with different cone angles, including upwind (negative) and downwind (positive) cone angles. With a fixed blade length, downwind, highly coned

rotors decrease the rotor swept area, resulting in both reduced power capture and blade loads. The load decrease is significant: 25 % compared with a 7 % decrease in power capture. In comparison with the length design study, it is clear why highly coned rotors are attractive for large rotor designs: an increased cone angle will decrease operational loads faster than an increase in blade length will increase them.

For all the aerodynamic design studies, there is a trade-off between power capture and blade loading. Each design study is

plotted together in Fig. 9, which also indicates the DELs in the flapwise and edgewise directions. In rotor design, our goal is to increase AEP and decrease blade loads, thus aiming to yield results in the lower-right quadrant of each plot.

The design changes can be applied in combination. If each individual design change, due to length, axial induction, or cone angle, is a vector in the (AEP, load) space of Fig. 9, the sum of those vectors are approximately the AEP and load of the new design. The SUMR-13B starts from the SUMR-13A, increases the available rotor power to 16.9 MW (blue diamond), decreases

the axial induction to 0.2 (red, dashed vector), and increases the cone angle to 12.5 deg (yellow, dashed vector). These design changes in combination result in the AEP and structural loading of the SUMR-13B: it increases AEP by 11 % compared to the SUMR-13A, while constraining peak blade flapwise loads to the level of the SUMR-13A. The increased blade length of the SUMR-13B increases the flapwise DELs due to the enhanced effect of wind shear and edgewise DELs due to the additional blade weight.

A set of three-bladed rotors (shown with dotted lines in Fig. 9) is designed similarly to the two-bladed design studies and exhibit similar trends to the two-bladed rotors in terms of blade loads. The blades of the three-bladed rotors experience lower loads with the same power capture due to their smaller chord and mass.

Despite the larger blade loads on 2-bladed rotors compared to 3-bladed rotors with the same power capture, we will be analyzing the 2-bladed SUMR-13B for the remainder of this article. When comparing similarly powered rotors, e.g. the CONR-

13 and the SUMR-13A, 2-bladed rotors reduce the total blade mass by as much as 25 %, which reduces the capital expenditures associated with blade material costs (Griffith, 2017). Given the constant AEP and decrease in CapEx of the 2-bladed rotors, we would expect the overall LCOE of a 2-bladed rotor to be less than that of a similarly powered 3-bladed rotor. However, periodic effects are more pronounced on the non-rotating components of 2-bladed rotors. We will analyze the load alleviating potential of different hub configurations in Sect. 9 and structural reinforcement in Sect. 8.2.





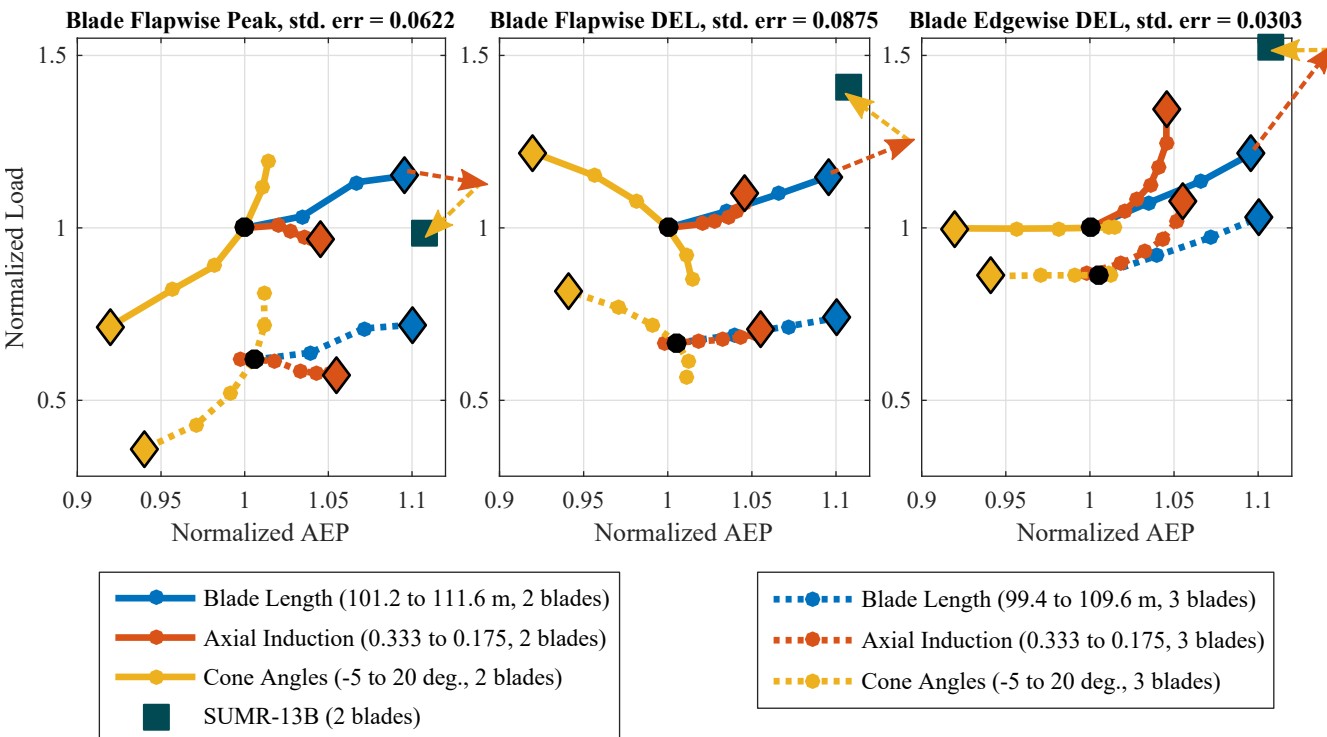

**Figure 9.** The trade off between power capture and blade loads. The AEP is plotted on the x-axis and blade loads are plotted on the y-axis. All rotors are normalized to the 2-bladed 101.2 m SUMR-13A baseline rotor design (black dot). Each dot represents a rotor design and each line represents the variation of one design parameter. Unless otherwise specified, the available rotor power is 13.9 MW, the axial induction is 0.333, and the cone angle is 5 deg.; the SUMR-13B is specified in Table 3. The normalized error standard deviation for AEP is the same as in Fig. 8 and the load error is normalized to the SUMR-13A loads. The vectors indicate design changes in combination: blade length increase (blue diamond), axial induction factor decrease (red, dashed vector), and cone angle increase (yellow, dashed vector) from the SUMR-13A to the SUMR-13B (square).

## 8.2 Blade structural parameters

As a wind turbine blade increases in length, its mass and stiffness increase to account for the additional structural loading. The structural properties of a blade are described by its distributed parameters along the blade span, which include mass-, stiffness-, and inertia-per-unit-length. In the previous section, these distributed structural parameters were constant for different blade lengths. Here, we will examine the effect of changing these parameters and, later, determine the mass and stiffness to model the larger SUMR-13B. To model blades with different lengths, we start with classical similarity scaling rules (Loth et al., 2017a), based on the length scaling factor

$$\eta = L/L_0, \tag{16}$$



where $L$ is the length of the scaled blade and $L_0$ is the length of the original blade. In this study, $L_0$ is the length of the baseline blades: the SUMR-13A for 2-bladed rotors and the CONR-13 for 3-bladed rotors. We will examine the scaling of the following parameters (Griffith and Ashwill, 2011):

- mass per-unit-length, which scales with $\eta^2$

- stiffness per-unit-length in the flapwise, edgewise, and torsional directions, which scales with $\eta^4$

- stiffness per-unit-length in the spanwise direction, which scales with $\eta^2$, and

- inertia per-unit-length in the flapwise and edgewise directions, which scales with $\eta^4$.

Once integrated over the blade length, e.g., the mass scales with $\eta^3$, while the stiffness and inertia properties scale with $\eta^5$.

These parameters can be more flexibly scaled to account for innovations or changes to the structural design. For instance,
we scale the mass-per-unit-length distribution by

$$M(r) = M_0(r)\eta^{2k_M}, \tag{17}$$

where $M(r)$ is mass-per-unit-length at spanwise location $r$ of the scaled blade, $M_0$ is the mass-per-unit-length of the original blade, and $k_M$ is a tunable parameter to increase or decrease the blade mass. Based on Eq. (17), a $k_M = 0$ would produce a blade with a mass that scales linearly with blade length, while $k_M = 1$ would produce a blade with a mass that scales with the
cube of blade length. State-of-the-art trends show that mass scales roughly with the square of blade length, or $k_M = 0.5$. A similar parameter can be defined for stiffness scaling

$$k_{s,flap} = k_{s,flap,0}\eta^{4k_{Fs}}, \tag{18}$$

where $k_{s,flap}$ is the flapwise stiffness-per-unit-length of the scaled blade, $k_{s,flap,0}$ is the flapwise stiffness-per-unit-length of the original blade and $k_{Fs}$ is a tunable flapwise scaling parameter. The edgewise stiffness will be similarly scaled using a
parameter $k_{Es}$. Flapwise and edgewise inertia will scale using the same mass scaling parameter $k_M$, but to the 4th power as in Eq. (18). Torsional and spanwise stiffness will scale according to the similarity scaling rules defined above, with $\eta^4$ and $\eta^2$, respectively. The SUMR-13B (2-bladed, $\eta = 1.24$) structural properties were scaled from the SUMR-13A blade, first separately each for the mass and stiffness parameters, and then all together (Full Scaling) in Fig. 10.

Ultimately the final structural parameters will be determined by the structural layup, but this model could be used to more
quickly analyze trade-offs between blade mass, stiffness, loads, and power. In general, mass scaling has the greatest impact on loads. Since this article only considers operational load cases, the effect is most apparent when analyzing fatigue loading. Loads during shutdown events and fault cases are also expected to increase with blade mass. Increased flapwise stiffness contributes to a small increase in energy capture (about 1 %, not shown) due to decreased blade deflection. We also observe that the change in load due to each individual scaling parameter ($k_M$, $k_{Fs}$, and $k_{Es}$) approximately sum (or combine linearly), when multiple
parameters are simultaneously scaled. This is shown in Fig. 10: the sum of the changes in load due to Mass, Flap. Stff., and Edge Stff., is approximately equal to the change in load due to Full Scaling. The same is true for the Final Design, which is a combination of the scaling parameters that are determined in the next section.





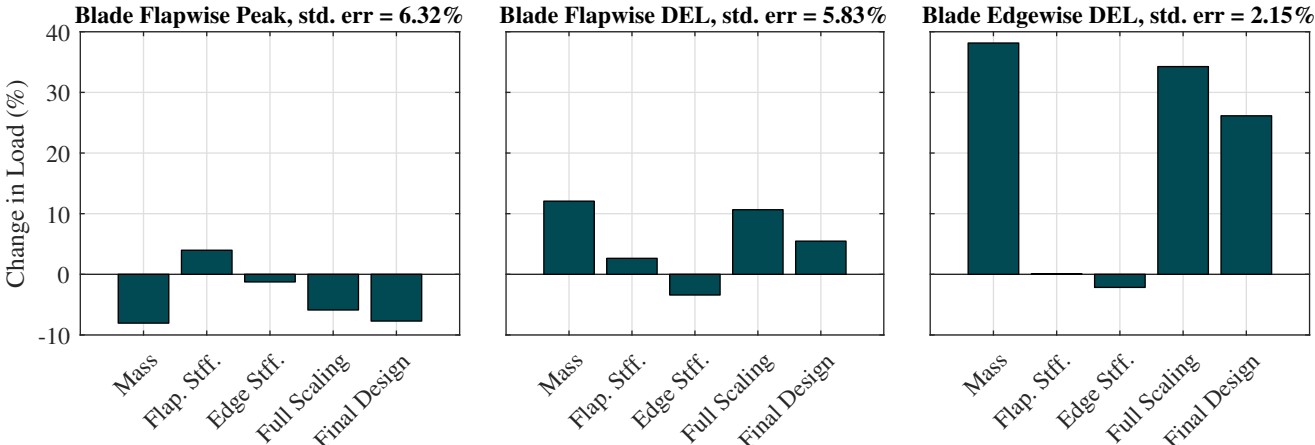

**Figure 10.** The effect of mass scaling ($k_M = 1$), flapwise stiffness scaling ($k_{Fs} = 1$), edgewise stiffness scaling ($k_{Es} = 1$), and the scaling of all parameters ($k_{All} = 1$) on blade loading compared with the non-scaled SUMR-13B ($k_{All} = 0$, which yield the SUMR-13B loads in Fig. 9). The standard deviation of error is computed using the calibration set in Table 3 and is normalized to the non-scaled SUMR-13B.

### 8.2.1 Selecting a $k_M$ and $k_{Es}$ for edgewise fatigue loads

The most significant impact of positive structural scaling is the increase in edgewise DELs due to the increased blade mass. Theoretically, the additional mass increase of the larger blade would provide additional reinforcement against these loads, through trailing edge reinforcement or increased root diameter. We see that changes to the blade mass impact the edgewise loads, i.e.,

$$\delta m_{bx} = a_1 k_M + b_1, \tag{19}$$

where $a_1$ and $b_1$ are determined from FAST simulations of the SUMR-13B blade with multiple $k_M$ values from 0 to 1 by finding the linear relationship between $k_M$ and $\delta m_{bx}$. Additional edgewise stiffness must compensate for the increase in edgewise load by increasing the ultimate load

$$m_{ult} = \frac{2\sigma EI_x}{c}, \tag{20}$$

where $\sigma$ is the fiberglass strain limit at the trailing edge, $EI_x$ is the edgewise stiffness, and $c$ is the blade chord. In terms of the scaling coefficients, a linearized version of Eq. (20) can be obtained

$$k_{Es} = a_2 \delta m_{bx} + b_2. \tag{21}$$

Finally, changes to the blade structural layup in the form of trailing edge reinforcement to increase edgewise stiffness will increase the blade mass

$$k_M = a_3 k_{Es} + b_3, \tag{22}$$

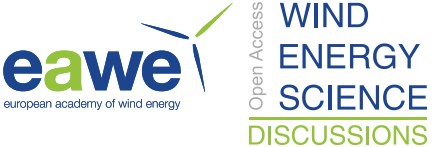

**Table 4.** Blade structural coefficients for the SUMR-13B blade.

| Structural Relations | | Final Design Coefficients |
| --- | --- | --- |
| $a_1 = 0.51$, | $b_1 = 1.40$ | $\delta m_{bx} = 1.8$ |
| $a_2 = 0.95$, | $b_2 = -0.79$ | $k_{Es} = 0.92$ |
| $a_3 = 0.87$, | $b_3 = 0$ | $k_M = 0.804$ |

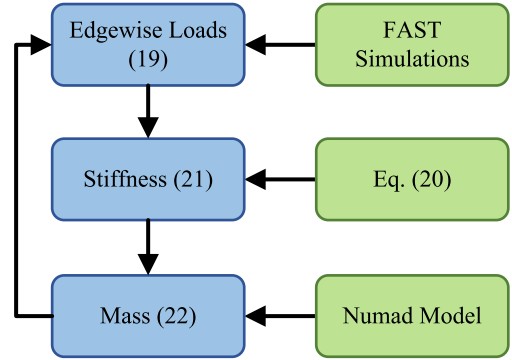

**Figure 11.** The relationship between blade mass, edgewise loads, and edgewise stiffness, as well how each value was derived.

where $a_3$ and $b_3$ are determined through a linear regression of multiple blade designs in NuMAD (Berg and Resor, 2012) with a target $k_{Es}$ from 0 to 1. Additional trailing edge reinforcement was applied to meet the target values within 5 % and the $k_M$ was computed using the overall mass of the resulting blade model.

The linear system determined by Eqs. (19), (21), and (22) can be solved to determine the necessary structural reinforcement for accommodating the load increase due to the increase in mass. See Table 4 for the results. These parameters can serve as targets for a detailed SUMR-13B structural layup design. For the remainder of this study, we will evaluate the loading on other components as a result of the mass increase shown in Table 4.

## 9 Hub configuration and main bearing loads

Blade loads are transferred through the blade root to the hub at the pitch actuator. In this section, we analyze the load cycle amplitudes of the hub loads and how they transfer to the non-rotating turbine components. The hub load axes, $y_h$ and $z_h$, rotate with the hub (Fig. 12). About the $y_h$-axis, hub loads are directly related to the blade loads for both 2- and 3-bladed configurations; they peak when the rotor is near $\psi = 0°$ due to vertical wind shear, resulting in a large cosine-cyclic component of the hub load about the $y_h$-axis ($m_{hy,c}^{1P}$). A teeter hinge reduces the coupling between blade and hub loads, except in cases of very large rotor deflections, where "hard" end stops increase the coupling and result in large peak loads. About the $z_h$-axis, the source of loading depends on whether the rotor has 2 or 3 blades (see Fig. 12). For 3-bladed rotors, the hub load about the $z_h$-axis is driven by the blade aerodynamic loading due to wind shear and has a similar magnitude to the load about the $y_h$-axis (Fig. 12, top right). This symmetry is not inherent in a 2-bladed configuration; the $m_{hz}$ load is primarily determined by the weight of the blades unless there is a horizontal wind shear. The mismatch between the load cycle amplitudes of $m_{hy}$ and $m_{hz}$ results in larger non-rotating loads, e.g., $m_{sy}$, for 2-bladed rotors (Fig. 12, bottom right). The hub load about the $z_h$-axis, for





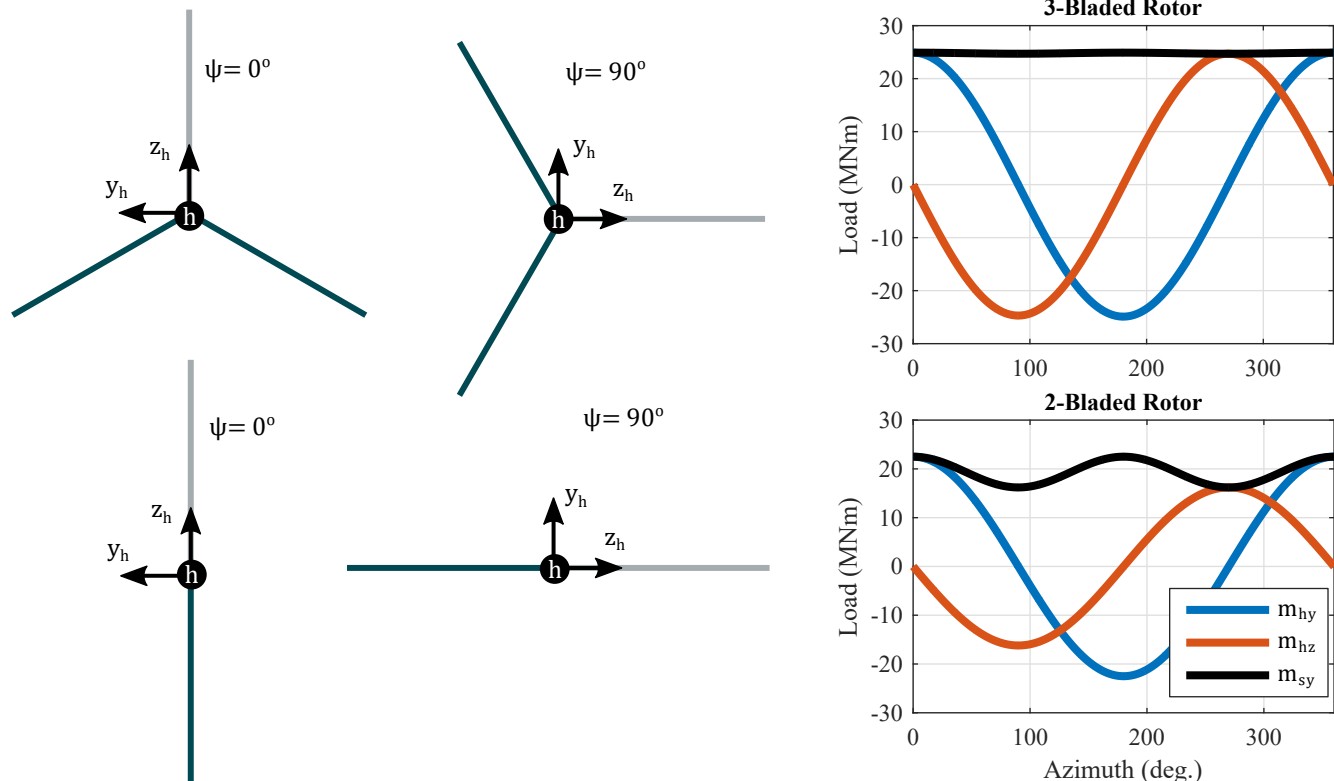

**Figure 12.** The hub axis (h) as it rotates with the rotor azimuth angle $\psi$ for a 3- and 2-bladed rotor. Note that the $y_s$-axis in Fig. 5 does not rotate while the $y_h$-axis in Fig. 12 does. An example timeseries of the hub loads ($m_{hy}$ and $m_{hz}$) is shown to demonstrate the difference in the non-rotating main bearing load ($m_{sy}$) for a 3-bladed (upper) and 2-bladed (lower) SUMR-13B rotor.

both hub configurations, peaks when the rotor is at $\psi = 90°$, resulting in a large $m_{hz,s}^{1P}$ component. The magnitude of these loads in relation to each other is important for determining their impact on the non-rotating load components.

The rotating hub is connected to the main shaft, which is supported by a main bearing close to the hub and also may consist of additional bearings between the hub and gearbox. A rotation matrix models the transfer of loads from the rotating to non-rotating frame

$$
\begin{bmatrix} m_{sy} \\ m_{sz} \end{bmatrix} = \begin{bmatrix} \cos\psi & -\sin\psi \\ \sin\psi & \cos\psi \end{bmatrix} \begin{bmatrix} m_{hy} \\ m_{hz} \end{bmatrix},
\tag{23}
$$

which results in the 1P hub loads mapping to a large 0P and 2P load component. The large 2P loads result in large fatigue DELs on the non-rotating parts of 2-bladed turbines. The hub configuration, including the number of blades, whether a teeter hinge is used, and IPC all have an impact on the fatigue loading of the main bearing.



**Table 5.** Comparison of the steady-state (8.5 ms$^{-1}$) hub load harmonics for 2-bladed fixed, teeter, and IPC methods, as well as 3-bladed (3b) rotors, in upwind and downwind positions. We analyze the cosine-cyclic hub load about the $y_h$-axis ($m_{hy,c}^{1P}$, Fig. 12) and the sine-cyclic hub load about the $z_h$-axis ($m_{hz,s}^{1P}$) because of their combined effect on non-rotating component loads. The different teeter and IPC methods are presented in Sect. 9.2.

| Rotor Location | Hub Configuration | Rotor Model | $m_{hy,c}^{1P}$ (kNm) | $m_{hz,s}^{1P}$ (kNm) |
|---|---|---|---|---|
| Downwind Rotors | 2-bladed Fixed Hub | SUMR-13A | 15500 | -8840 |
| | | SUMR-13B | 22500 | -16200 |
| | 2-bladed Teeter | Free Teeter | 0 | -16900 |
| | | Ideal Teeter | 16200 | -16400 |
| | 2-bladed IPC | Blade IPC | 12300 | -16200 |
| | | Bearing IPC | 17700 | -16200 |
| | 3-bladed Fixed Hub | SUMR-13A (3b) | 7180 | -7220 |
| | | SUMR-13B | 24900 | -24700 |
| Upwind Rotors | 2-bladed Fixed Hub | SUMR-13A | 3780 | -3570 |
| | 3-bladed Fixed Hub | SUMR-13A (3b) | -526 | 543 |

## 9.1 Number of blades

To compare with the 2-bladed SUMR-13B, a 3-bladed SUMR-13B was designed using the same blade parameters described in Table 4. Peak and fatigue blade loads in both the flapwise and edgewise directions are unaffected by the change in the number of blades.

Loads on other turbine parts are, however, affected by the change in the number of blades. Hub loads on the 2-bladed SUMR-13B are mostly about the $y_h$-axis (see $m_{hy,c}^{1P}$ in Table 5), while 3-bladed rotors are balanced in both directions. The hub loads in Table 5 can be mapped to the non-rotating frame by Eq. (23). The 1P harmonic in the rotating frame transfers to 0P and 2P harmonics according to

$$m_{sy}^{0P} \quad = \tfrac{1}{2}(m_{hy,c}^{1P} - m_{hz,s}^{1P}) \tag{24}$$

$$m_{sy}^{2P} \quad = \tfrac{1}{2}(m_{hy,c}^{1P} + m_{hz,s}^{1P}). \tag{25}$$

The 3P component is determined similarly based on the 2P harmonic load components.

    Three-bladed rotors are advantageous due to these balanced hub loads, which effectively nullify the 2P load components and only contain a small 3P load on the non-rotating turbine components. The difference in magnitude of the 1P hub load harmonics is responsible for the greater loading on the non-rotating components of 2-bladed rotors. Figure 13 shows more than

a 20 % reduction in main bearing DEL for the 3-bladed SUMR-13B, compared to the 2-bladed, fixed-hub SUMR-13B, even with a significantly more powerful rotor.

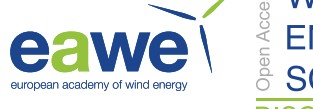
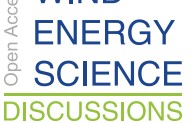


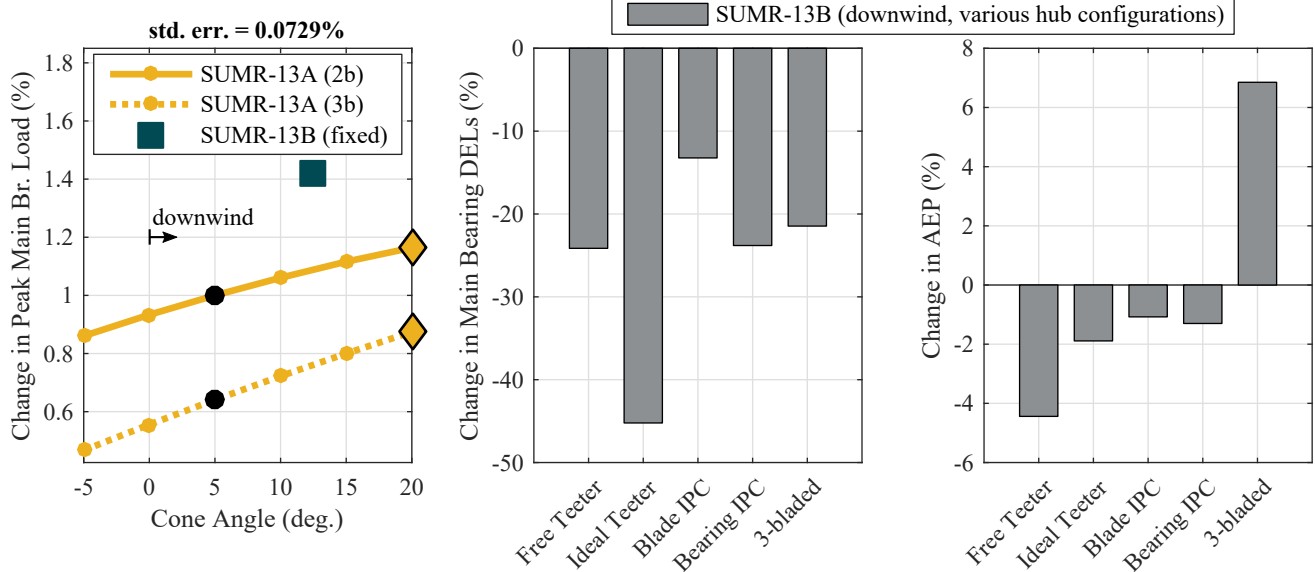

**Figure 13.** Change in peak main bearing loads (left) for SUMR-13A cone angle study (2- and 3-bladed rotors) and SUMR-13B fixed hub configuration, change in main bearing DELs (middle) about the $y_s$-axis (DELs about the $z_s$-axis are within 5 % of the $y_s$-axis DELs) and change in AEP (right) for various hub configurations of the SUMR-13B, compared with the fixed-hub, 2-bladed SUMR-13B Final Design described in Sect. 8.2. The DEL and AEP results from different hub configurations (center and right) are design loads computed directly from DLC simulations.

## 9.2 Teeter and individual pitch control

Historically, 2-bladed turbines have used a mechanical teeter hinge, which allows for rotation about an axis perpendicular to the main shaft at the shaft tip. Recently, with the advent of pitch regulated turbines, individual pitch controllers have been designed in order to mimic this action by changing the aerodynamic loads on the blades as they rotate. Both solutions reduce
loading on the hub, which translates into reduced loading on the main bearing and other non-rotating components.

We have modeled a free teeter hinge in FAST by enabling the teeter degree-of-freedom and setting a zero damping coefficient to the teeter motion. This free teeter setup would provide the best configuration for reducing blades loads. A more realistic teeter hinge must account for friction, damping, and end stops (see, e.g., Schorbach et al. (2017)).

The free teeter hinge configuration completely eliminates the coupling between blade and hub loads, resulting in zero hub
loads about the $y_h$-axis. The relationship in Eq. (25) and steady-state results (Table 5) suggest that main bearing fatigue loads $(m_{sy}^{2P})$ increase when compared to the fixed hub configuration. However, DLC simulations show that turbulence has a relatively minimal impact on the non-rotating components for this rotor with a free teeter hinge, compared with all other rotors. In other words, the design loads for the main bearing are nearly equal to the steady-state loads, but in every other case there is a significant turbulent component, as mentioned in Sect. 6. For this reason, it is omitted from the calibration set of 2-bladed
rotors. Instead of presenting the calibrated harmonic load estimates and power capture, we present the design loads computed



directly from DLC simulations in Fig. 13. However, the steady-state results in Table 5 still illustrate how a more optimal teeter design could mimic the balanced hub loads of 3-bladed rotors.

A more ideal teeter design could be achieved by selecting an appropriate teeter damping coefficient $d_{\text{teet}}$ that matches the $m_{hy,c}^{1P}$ and $m_{hz,s}^{1P}$ load harmonics to minimize the main bearing load $m_{sy}^{2P}$. Since only one damping coefficient must be designed for all wind speeds, we minimize the main bearing load using the wind speed distribution $p(u)$ by

$$d_{\text{teet,opt}} = \operatorname*{arg\,min}_{d_{\text{teet}}} \sum_{u \in U_{\text{teet}}} p(u) m_{sy}^{2P}, \tag{26}$$

where $U_{\text{teet}}$ is the set of wind speeds used to analyze the teeter damping, focused on below-rated operation, where the greatest fatigue contribution occurs. Main bearing load cycle amplitudes $m_{sy}^{2P}$ and $m_{sz}^{2P}$ increase with wind speed due to the increased effect of wind shear, but lower wind speeds are far more probable than high wind speeds. Since our design goal is to reduce fatigue loads on the main bearing and other non-rotating components, we focus on below-rated wind conditions. The ideal teeter design greatly reduces the main bearing fatigue loads, along with the fatigue loading on the other non-rotating components, but reduces energy capture by 1.9 %, compared with the fixed 2-bladed SUMR-13B (Fig. 13, center and right).

Alternatively, IPC can be used to mimic the rotor balancing of a teeter hinge by adding a time-varying pitch angle offset to each blade. An IPC algorithm was initially designed to focus on blade loads, which we call Blade IPC in Table 5 and Fig. 13. The 2-bladed IPC architecture used here was initially presented in van Solingen and van Wingerden (2015), which minimizes the teeter load

$$m_{\text{teet}} = \frac{1}{2}(m_{by,1} - m_{by,2}). \tag{27}$$

We have applied loop-shaping procedures (McFarlane and Glover, 1992) to fine tune the controller to reduce the 1P and 2P blade harmonics, which results in a decrease in the blade design load for the SUMR-13B (about 10 % for flapwise peak and fatigue loads). The IPC algorithm was designed to operate both above- and below-rated, since the bulk of the fatigue loads occur below rated, and the IPC must be active near rated in order to reduce the peak design load. Since this Blade IPC is designed to reduce blade loads as much as possible, hub loads about the $y_h$-axis are less than hub loads about the $z_h$-axis (Table 5). Therefore, the Blade IPC algorithm is not necessarily optimal for the main bearing DELs.

Using the relationship in Eq. (25), we designed a Bearing IPC algorithm with the goal of balancing the hub load components, such that $m_{hy,c}^{1P} = -m_{hz,s}^{1P}$, to minimize 2P loading on the main bearing. Equivalently, $m_{hy}^{1P}$ and $m_{hz}^{1P}$ should be equal in magnitude and 90° out of phase. Since $|m_{hz}^{1P}|$ changes more slowly than $|m_{hy}^{1P}|$, the $m_{hz}$ signal is delayed by 90° and the difference

$$m_d = m_{hy} - m_{hz}(\psi - 90°) \tag{28}$$

can be fed back using the same architecture as the Blade IPC because $m_{hy} = 2m_{\text{teet}}$. Harmonic load estimates suggest better load mitigation than those in Fig. 13, so we present the DLC-based design loads directly from turbulent simulations. In general, dynamic control solutions are not as well estimated using harmonic load estimates, compared with changes to the rotor model



using the same control. Other control methods were attempted to balance the load components in Eq. (25), which are further explored in Zalkind and Pao (2019).

If used below rated, these load mitigation techniques reduce power capture, as shown in the right of Fig. 13. IPC can be designed so that it only operates above rated, resulting in a negligible power loss. However, this reduces its effectiveness in 5 constraining peak loads that occur close to rated wind speeds.

## 9.3 Large cone angle effects

The main bearing must support the weight of the rotor and thrust imbalance on the rotor due to shear, i.e.

$$m_{sy}^0 = m_{sy,\text{grav}}^0 + m_{sy,\text{shr}}^0. \tag{29}$$

For downwind turbines, both components of Eq. (29) are positive, resulting in large, constant main bearing loads about the 10 $y_s$-axis. For upwind turbines, the load due to gravity $m_{sy,\text{grav}}^0$ is negative while the load due to wind shear $m_{sy,\text{shr}}^0$ is positive, which greatly reduces the steady-state main bearing load for upwind turbines compared to downwind turbines. To quantify this difference, we analyze the calibrated harmonic load estimates of the peak main bearing load ($m_{sy}^{\text{Peak}} = m_{sy}^{0P} + m_{sy}^{2P}$) for rotors with various cone angles (Fig. 13, left).

The steady-state results in Table 5 suggest there would be a significant change in the mean main bearing load $m_{sy}^{0P}$ going from 15 upwind to downwind rotor configurations. However, the design loads computed using DLC simulations show that turbulence contributes a large amount to the peak load experienced by the main bearing (Fig. 6) for both configurations. A downwind configuration, compared to the same rotor upwind (with cone angles of $\pm 5$ deg., respectively) only increases the main bearing load by about 15 %. Despite the larger total blade mass of the 3-bladed rotors, 2-bladed rotors still have a larger peak load due to the increased 2P loading and a larger turbulent load component. We see this same effect in the fatigue loading results 20 of Fig. 13, which suggests that main bearing peak loads could be reduced using the same methods as in Sect. 9.2. The larger SUMR-13B, however, has a non-negligible increase in the peak main bearing load, due to combined increases in blade mass, blade length, and cone angle. These increased loads on the main bearing transfer to the other non-rotating components, which we will analyze in the yaw bearing and tower design studies.

## 10 Yaw bearing loads and nacelle layout

The main bearing is mounted to the bedplate of the nacelle, which attaches to the yaw bearing, responsible for rotating the entire nacelle and rotor to align with the wind direction. The yaw bearing experiences similar loads to the main bearing; they peak near rated and at cut-out due to thrust effects and wind shear, respectively. A potential issue with downwind turbines is a large, mean $y_y$-axis moment leading to large peak yaw bearing loads, similar to the peak main bearing load. However, peak loads on the yaw bearing can be counteracted by properly balancing the nacelle center-of-mass atop the tower. We will study 30 the different cone angle designs from Sect. 8.1 for 2- and 3-bladed rotors, as well as our SUMR-13B Final Design to investigate the effect of rotor cone angle and increased mass on nacelle design and yaw bearing loads.



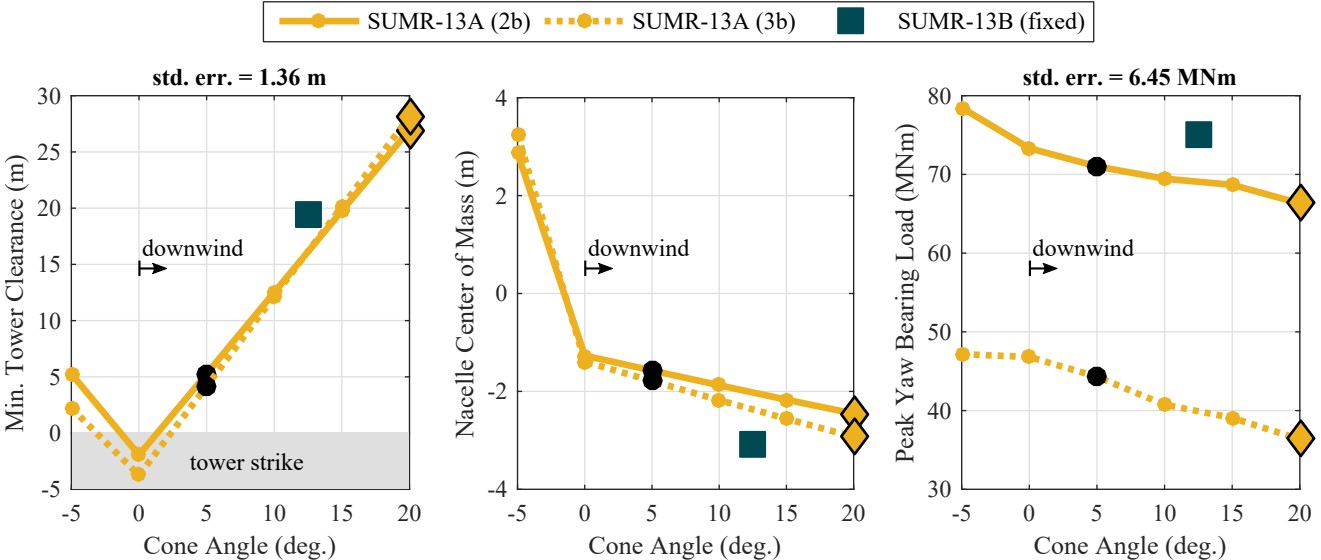

**Figure 14.** The tower clearance (left) resulting from upwind (negative cone angles) and downwind (positive cone angles) configurations, the nacelle center of mass (middle) required to balance the rotors, and the peak yaw bearing loads (right) of the balanced rotors.

Large mean loads on the yaw bearing $m_{yy}^{0P}$ cause large peak loads that can be overcome by properly choosing the hub-to-tower overhang $x_{OH}$ and the nacelle center of mass $x_{cm}$ (as shown in Fig. 5). We use a simple method for determining the nacelle overhang: for upwind turbines, the nacelle overhang was set to that of the CONR-13 (-8.61 m), and for downwind turbines, we used the minimum possible overhang (3.15 m, equal to the radius of the tower at the nacelle). These hub-to-tower

overhang values result in adequate tower clearance when the cone angle is at least $5°$ in either direction (Fig. 14, left). Rotors with larger cone angles have large tower clearances, which is part of the motivation for their design.

To compare peak yaw bearing loads across rotors, we adjust the nacelle center of mass so that mean yaw bearing loads $m_{yy}^{0P}$ are minimized in 0 ms$^{-1}$ winds. The mean yaw bearing load is linearly dependent on the component masses and center-of-masses

$$m_{yy}^{0P} = g(m_{nac}x_{cm} + m_{rot}x_{cm,rot}),\qquad(30)$$

where $g$ is the acceleration due to gravity, $m_{nac}$ is the nacelle mass, $m_{rot}$ is the total rotor mass, and $x_{cm,rot}$ is the rotor center of mass. The nacelle center of mass $x_{cm}$ that minimizes the mean overturning yaw bearing load is

$$x_{cm} = -\frac{m_{rot}x_{cm,rot}}{m_{nac}}.\qquad(31)$$

The hub and nacelle masses are scaled using a length-to-mass scaling factor of $\left(\frac{100}{63}\right)^3$ from the NREL 5 MW reference

turbine (Jonkman et al., 2009), and shown in Table 6. The hub and nacelle masses are constant for all rotors throughout this study, but the rotor mass and center-of-mass vary.



**Table 6.** Component masses for placing the nacelle center-of-mass atop the tower.

| Component | Mass (Mg) |
|---|---|
| Nacelle | 1030 |
| Hub | 245 |
| Blade (2-bladed SUMR-13A) | 51.8 |
| Blade (3-bladed SUMR-13A) | 47.3 |
| Blade (2-bladed SUMR-13B) | 83.8 |

Rotors with large downwind cone angles must have nacelle center-of-masses further upwind (negative values in Fig. 14, center). Given the nacelle mass in Table 6, moving the nacelle center of mass 1 m upwind reduces the mean (and peak) yaw moment by about 10 MNm. Due to the extra overhang necessary for upwind turbines, the center of mass location for the downwind turbines is closer to the tower than for the upwind turbines. By designing the proper hub to tower overhang and nacelle placement, the peak yaw loads are no more problematic for downwind rotors than upwind rotors. Once properly balanced, the peak yaw loads are primarily driven by the thrust imbalance due to wind shear, which decreases with increased cone angle (Fig. 14, right). However, changing the nacelle center of mass is a non-trivial task that involves a detailed drivetrain and nacelle design. Fatigue loads (not shown) on the yaw bearing also depend on rotor thrust and decrease with increasing cone angles. The methods presented in Sect. 9 also reduce yaw bearing loads.

## 11   Tower loads

The yaw bearing is attached to the top of the tower, which must support the rotor-nacelle assembly and withstand large moments. We focus on the effect of rotor axial induction, cone angle, and the number of blades on peak loads in the fore-aft direction $m_{ty}^{\text{Peak}}$ and fatigue loading in the side-to-side direction $m_{tx}^{\text{DEL}}$.

Peak fore-aft tower loading is similar to the peak blade loads described in Sect. 8.1; with a maximum near rated wind speeds, they are largely driven by rotor thrust, which is most sensitive to changes in axial induction and cone angle. Lower axial induction rotors and downwind rotors can both reduce the peak tower load by as much as 20 % (Fig. 15, left). Tower loads are not as sensitive to blade length. Longer blades increase the rotor thrust in below-rated wind speeds, but with a constant generator power, the pitch controller activates at lower wind speeds, constraining the peak tower load near rated. For rotors that capture the same amount of power, 2-bladed rotors experience about a 30 % increase in peak tower fore-aft load when compared to 3-bladed rotors, primarily due to a large difference in the turbulent sampling of the wind, an effect also present when looking at the tower DELs.

Besides having larger chord lengths that sample more turbulence than 3-bladed rotors, 2-bladed rotors also experience a resonance due to the tower design. Modern wind turbine towers are usually designed to be "soft-stiff", with a natural frequency between the 1P and 3P harmonic of the rotor (van der Tempel and Molenaar, 2003). When the 2P rotor speed interacts with



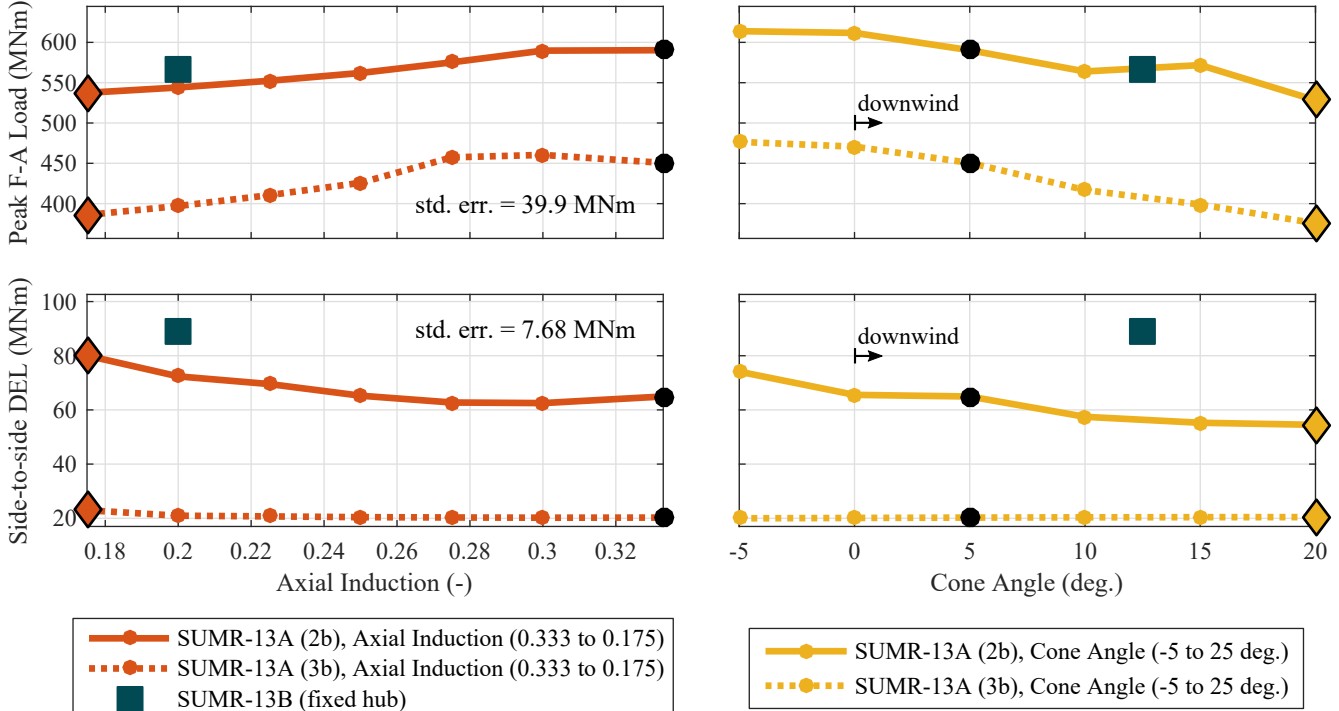

**Figure 15.** Peak tower loads in the fore-aft (F-A) direction ($m_{ty}^{\text{Peak}}$) and side-to-side DELs ($m_{tx}^{\text{DEL}}$) for rotors with different axial induction factors (red), cone angles (yellow), and number of blades. The same loads for the SUMR-13B are also shown. Unless otherwise specified, the available rotor power is 13.9 MW, the axial induction is 0.333, and the cone angle is 5 deg.; the SUMR-13B is specified in Table 3. The standard deviation of error for both load axes incorporates all of the presented design studies.

the natural frequency of the tower, there are high fore-aft and side-to-side loads. Side-to-side tower DELs increase the most, since there is less aerodynamic damping from the rotor in this direction (Jonkman and Matha, 2011). One idea is to use a high-compliance tower structure (Bergami et al., 2014) or a floating substructure with a natural frequency below the 1P harmonic. However, a very low tower natural frequency causes tower motion to be perceived as a wind speed disturbance, resulting in

5  speed regulation issues. Several studies have considered this, given the emergence of floating wind turbines (Jonkman and Matha, 2011), but to simplify our analysis, we have kept the same tower for all turbines: a scaled version of the NREL-5MW 3-bladed reference model (Jonkman et al., 2009).

Our solution is to implement a speed avoidance controller that reduces the rotor speed as it approaches the critical rotor speed from below and increases it after, avoiding the critical speed as much as possible (Fig. 2, center). Similar approaches

10  have been used in 2-bladed rotor field testing (Johnson et al., 2005). While this controller does reduce side-to-side fatigue loading, 2-bladed rotors still experience 3–4 times the DELs that similar 3-bladed rotors experience (Fig. 15). Longer, heavier blades with lower axial induction factors amplify this effect. Changing hub architectures also impacts the tower fatigue loads. Both teeter and IPC decrease the fore-aft loading while increasing the side-to-side loading.



Steady-state simulations predict the same peak tower loads for both 2- and 3-bladed rotors, but turbulent simulations show a clear difference in the design load, as indicated in Fig. 15. Compared with other turbine parts, the calibrated estimates of the tower loads have a large amount of uncertainty (Fig. 6). This uncertainty can be attributed to the source of these tower loads, which are highly dependent on turbulent gusts.

## 12   Model limitations, suggested improvements, and potential use

When analyzing the design studies of Sects. 8–11, we have come across a few sources of uncertainty in the calibrated harmonic load estimates. When calibrating the quasi-steady harmonic estimates to loads calculated using DLCs (Sect. 6), we see that a large component of the design load is due to turbulence, which primarily depends on the number of blades on the rotor, leading to different calibration coefficients for 2- and 3-bladed rotors in Eq. (14). However, the turbulent component is also correlated with other model parameters, most notably rotor thrust. Highly coned downwind rotors reduce the rotor thrust and have a lower turbulent component than upwind rotors. Additionally, dynamic effects, like the problematic gust in Fig. 3 are not explicitly modeled in the steady-state harmonic model of Sect. 5. Thus, dynamic control solutions that appear promising in steady state should be ultimately verified in turbulent simulations.

Several improvements to the steady-state harmonic model could be made. For instance, the problematic gust events follow a similar profile in many instances; this could be an additional simulation added to the model's set of simulations. While outside the scope of this study, parked, fault, and shutdown cases can result in the largest design loads in practice, e.g., in (Griffith and Richards, 2014); they could be added with little computational expense. The calibration procedure could be streamlined by perhaps doing a single, exemplary turbulent simulation for each case to determine the turbulent component of each load.

The calibrated harmonic load estimation method used to evaluate design trade-offs for energy capture and design loads presented in this article provides a potential middle ground for wind turbine system engineering tools. The method is more realistic than simple scaling rules and static estimates, but requires less computational effort than full sets of DLC simulations and therefore allows for an initial optimization over a wider range of configurations.

## 13   Conclusions

In this article, we presented a method for estimating wind turbine power capture and structural loads, which uses the harmonic components of quasi-steady aeroelastic simulations in FAST. The power and load estimates are calibrated against the power producing design load cases and could be used for initial wind turbine system design or sensitivity analyses to model changes. We designed 42 different rotors with the goal of reducing the cost of wind energy through increased power capture and reduced capital expenditures. Power capture and structural loads are analyzed for blades longer than 100 m in both upwind and downwind configurations, with 2- and 3-bladed rotors, leading to an updated design, the SUMR-13B, with longer, more slender blades that align with industry trends. A series of detailed design studies was performed, with the following conclusions.



- Low axial induction rotors using longer blades with smaller chord lengths can capture more energy while constraining peak operational blade loads.

- As rotor size increases, due to increasing blade mass, edgewise blade loading becomes a critical design-driving load and may ultimately constrain the size of wind turbine rotors.

- Downwind, coned rotors can significantly reduce peak operational blade loads, but capture less energy than rotors with lower cone angles.

- Downwind, coned rotors will experience slightly larger (about 15-25 %) peak main bearing loads than upwind turbines, but the effect is amplified with increasing blade length, mass, and cone angle.

- Peak yaw bearing and tower loads are not problematic for downwind rotors as long as the nacelle is properly balanced on the tower.

- 2-bladed rotors experience significantly greater non-rotating loads compared to 3-bladed rotors, unless a teeter hinge or individual pitch control is utilized. In these cases, the loading is comparable, but with a loss in power.

- 2-bladed rotors will require either speed avoidance control or a different tower design to avoid resonance with the 2P frequency of the rotor.

We believe that our model has provided future wind turbine designers with a method for more quickly analyzing design trade-offs, and our design studies can serve as a reference for future large rotor designs.

*Data availability.* The data from this study can be made available upon request.

*Competing interests.* The authors declare no competing interests.

*Acknowledgements.* The information, data, or work presented herein was funded in part by the Advanced Research Projects Agency - Energy (ARPA-E), U.S. Department of Energy, under Award Number DE-AR0000667. The views and opinions of authors expressed herein do not necessarily state or reflect those of the United States Government or any agency thereof.





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
