# Peer review of "System-level design studies for large rotors"

_Wind Energy Science, 2018_

## Referee Comment (RC1) · Christopher Kelley (Referee) · 25 Apr 2019

Overall the idea of reducing the complexity of design load case simulations using the harmonic model proposed has merit. I agree with the authors that it is very useful to examine how the effect of design parameters such as cone angle, blade length, and axial induction have on loads all with reduced computational cost. I have a few questions for the authors that I would like to see addressed in the final version of the journal article.

1. By considering the rotors bending motion only a function of azimuthal angle, are you ignoring the fact that resonance of the structure may be uncorrelated with azimuth? In other words, the structures flapping due to resonance may sometimes align with a specific azimuthal angle on one revolution, but not another? I think this is a source of confusion for me since I am not as familiar with this harmonic analysis. But maybe

this is the key simplification to reduce computational cost, as opposed to letting random blade motion and turbulence appear with long timeseries like in FAST. A further explanation would be useful.

2. In Section 4, when discussing the closed loop controller, it would be good to describe what pitch rate was the outcome of the gains for the PI controller to make sure the maximum blade pitch rate is physically possible. For a 13 m blade 5-10 deg/s is reasonable, but for a 13 MW blade 1-3 deg/sec would be realistic. This can drastically change the 50 year DLC 1.1 result.

3. In equation 11, is m_ss for steady state amplitude equivalent to the 0th order amplitude, m_0?

4. For Figure 6, I think a further explanation of interpreting the turbulence factor and std error/mean would be helpful. Is f_turb indicative of the mean error between the harmonic model and the FAST simulations? And is std error/mean indicative of the average dynamic error?

5. In equation 13, does this mean you need two calibration constants for each of the 3 azimuthal modes you are considering?

Figure 9 seems to be showing a lot of interesting trends. It might be useful to inform the reader which design load cases were the driving cases. For example, increasing damage equivalent load but decreasing maximum peak load might be ok if tip deflection is the driving DLC.

Most of my questions are just about helping the reader and me better understand the implications of the harmonic analysis described. Thank you for consideration of my questions and comments and your interesting work to improve blade design.

---

## Referee Comment (RC2) · Anonymous Referee #2 · 9 Jun 2019

The aim of the present work is to examine the effect of rotor design choices on the power capture and structural loading of each major wind turbine component. The authors present an interesting approach towards the analysis of the impact of design parameters such as cone angle, blade length, and axial induction. Some major clarifications are needed before it can be published. Some specific comments are listed below.
* * *
1. In section 3, it is unclear what is the justification of assuming that the tower fairing is present in the current simulations. Can the authors be elaborated.

2. In section 4, do the authors have an explanation why at different wind speeds, the rotors have different transitions? How's that can affect the parameters of control architecture?

3. In section 5, why the authors select only the first four harmonics to reconstruct the

load signal? is that because of the linearity of these harmonics or because of their energy? How would the results change if we consider more or random harmonics?

4. In subsection 5.1, the author used Weibull distribution which is one of the most widely used lifetime distributions in reliability engineering, can the authors explain what is the range of Whöler exponent used in the steady state and with turbulence cases? Can the authors explain the effect of the gust or rare-events on the Whöler exponents and the dominant harmonic components.

5. In section 6, it is unclear why the peak loads are more deterministic than DELs and rotating component loads are more deterministic than non-rotating component loads?

6. In section 6, the authors mentioned that "turbine parts have large turbulent components that are not directly modeled in steady state"; what are the authors mean by not directly modeled, and how much this point can affect the correlation and standard deviation shown in figure 6?

7. In section 6, do the linear calibration factors are related to selecting only first four harmonics?

8. In section 6, what is the correlation between the uncertainty and level of turbulence?

9. In section 6, the authors mentioned that "the most erroneous value (using the metric of standard deviation normalized by the mean) is the minimum tower clearance, but this is influenced by the small average tower clearance over all rotors" How much this point affect the result by assuming the tower fairing is present?

10. In subsection 8.1, can the authors be elaborated on the impact of the constraining the transition on the above-rated control at lower wind speeds?

11. In subsection 8.1, I see a potential innovation in the analysis and comparison should be provided in figure 9.

12. In subsection 8.1, the authors highlighted that the blades of the three-bladed rotors
experience lower loads in comparison with the two-bladed rotors, it is unclear if this is the case for the damage equivalent load, can the authors explain that.

13. In subsection 9.1, please can the authors explain how 2P harmonic load components were used to determine 3P load. Also, what is the impact of the 2P and 3P on rotating and non-rotating parts? Did this analysis include the effect of steady-state and turbulence? Please be elaborated.

14. In section 9.2, it is unclear why the authors omitted the turbulent component from the calibration set of 2-bladed rotors.

15. In subsection 9.3, can the authors highlight the turbulence contribution on the peak load on the main bearing and the other non-rotating components in term of the cone angle and combine that with the yaw bearing and tower.

16. Can the authors include the citations for the equations used in this study?
* * *
In closing, thank you for considering my comments. I found this work potentially interested and could help in improving blade design.

---

## Author Response (AR1)

**Response to Referee #1**

**MS Number:** wes-2018-80
**Title:** System-level design studies for large rotors
**Corresponding author:** Daniel Zalkind

We would like to thank referee Christopher Kelley for his review and comments on our research paper. Through a major revision of the manuscript (including a formal internal review by the National Renewable Energy Laboratory), we have tried to address all the referee's comments and we feel the result is a much stronger contribution. The following table collects the referee's comments, the authors' responses to each point, and the authors' changes in the manuscript. In addition, a color-coded version of the manuscript is provided, in which all changes can be easily identified. We have used the red color to indicate text that has been removed from the submitted manuscript. The descriptions in blue represent the added or re-written parts, addressing the referee's comments.

| Comments of Referee #1 | Authors' Responses |
|---|---|
| 1. By considering the rotors bending motion only a function of azimuthal angle, are you ignoring the fact that resonance of the structure may be uncorrelated with azimuth? In other words, the structures flapping due to resonance may sometimes align with a specific azimuthal angle on one revolution, but not another? I think this is a source of confusion for me since I am not as familiar with this harmonic analysis. But maybe this is the key simplification to reduce computational cost, as opposed to letting random blade motion and turbulence appear with long timeseries like in FAST. A further explanation would be useful. | **Answer:** Yes, this is the key simplification to reduce the computational expense associated with performing the full set of simulations necessary to find design loads. We found that only considering the loads due to wind shear and turbine self-weight provides a reasonable representation of the loads for the objective of comparing different turbines. For example, turbines with larger blades will experience a larger 1P blade load and will also experience a similar increase in loading due to turbulence and non-periodic loads. These non-periodic components are not modelled by the transformation and are instead considered as part of the turbulent component of the load in Section 6. |
| | **Changes in manuscript:** When introducing the harmonic model, the simplification and use for the harmonic loads is explained: |
| | In this section, we describe harmonic loads $m^H$, which are derived from constant and periodic loads that arise due to steady wind loading, wind shear, and turbine self-weight. These harmonic loads can be mapped, or transformed, into estimates $m^{Est}$ of design loads $m^{DLC}$ that are computed using operational DLC simulations in Sect. 6. The key simplification of the harmonic load model compared to design loads computed using DLC simulations is the omission of load variations that do not correspond to the rotor speed. These non-harmonic load variations arise because of |

wind speed and direction changes, as well as the component's natural frequencies. All frequency components are necessary to determine the design load for a final, detailed design, but for exploring potentially large numbers of design trade-offs, simplified harmonic loads provide enough information about the various turbine loads.

It is clarified again when discussing the derivation of peak and fatigue loads from the harmonic components of simulations with a constant, sheared inflow:

The loads at higher harmonic and natural frequencies contribute to both fatigue and extreme loads, but since our goal is to derive a mapping from a simplified computation (harmonic load) to a more expensive simulation (design load), their effects are neglected and considered as part of the uncertainty of the transformation in Section 6.

Finally, it is mentioned in Section 5.2 (Harmonic versus turbulent loads) that non-periodic loads are not modelled in the transformation from harmonic to design loads:

The structural loads on a wind turbine originate from  constant and periodic effects, modeled by the harmonic load, as well as from dynamics due to turbulence , which are not necessarily correlated with the azimuthal position of the rotor and are not modeled in this transformation.

| | |
|---|---|
| 2. In Section 4, when discussing the closed loop controller, it would be good to describe what pitch rate was the outcome of the gains for the PI controller to make sure the maximum blade pitch rate is physically possible. For a 13 m blade 5-10 deg/s is reasonable, but for a 13 MW blade 1-3 deg/sec would be realistic. This can drastically change the 50-year DLC 1.1 result. | **Answer:** During turbulent simulations (DLCs 1.2 and 1.3), the maximum pitch rate for the SUMR-13A is 2.45 deg/sec and 2.18 deg/sec for the SUMR-13B. During the extreme coherent gust with direction change (DLC 1.4), the maximum pitch rate limit of 4 deg/sec is not violated for either the SUMR-13A or SUMR-13B. For comparison, the NREL-5MW reference turbine (with 63-meter-long blades) has a maximum pitch rate limit of 8 deg/sec |
| | **Changes in manuscript:** A sentence on the pitch actuator rates is added in Section 4: |
| | The pitch actuator has a maximum pitch rate limit of 4 $°s^{-1}$; turbulent simulations have a maximum pitch rates between 1 and 3 $°s^{-1}$ were recorded in the turbulent simulations that were run. |

| | |
|---|---|
| 3. In equation 11, is $m^{SS}$ for steady state amplitude equivalent to the 0th order amplitude, $m^0$? | **Answer:** Thank you for raising this question; these terms can be confusing. In eq. (11) we are referring to the harmonic load (peak or fatigue) that is derived from the mean load $m^0$ and dominant harmonic load component $m^{nP}$ across wind speeds. The harmonic load is used as a surrogate model to estimate the design loads that are computed from DLC simulations. |
| | **Changes in manuscript:** Throughout the article we have eliminated the usage of "steady" and "quasi-steady" to use more precise language when describing how the load was generated, e.g., using harmonic loads $m^H$, we derive estimated loads $m^{Est}$ that should approximate loads computed from DLC simulations $m^{\mathrm{DLC}}$ with some residual. |
| 4. For Figure 6, I think a further explanation of interpreting the turbulence factor and std error/mean would be helpful. Is $f^{turb}$ indicative of the mean error between the harmonic model and the FAST simulations? And is std error/mean indicative of the average dynamic error? | **Answer:** Thank you for this comment. $f^{turb}$ is used to indicate how much of the load can be attributed to turbulence versus steady and periodic effects (harmonic load). For example, the mean harmonic ($m^H$) peak main bearing load about the y-axis is approximately 10 MNm, the mean DLC ($m^{DLC}$) peak main bearing load is approximately 40 MNm. Thus, we say the mean turbulent ($m^{turb}$) load is approximately 30 MNm, using the definition in (11), and the turbulence factor $f^{turb}$ is 0.75. |
| | Std. error/mean is not indicative of the average dynamic error between the harmonic and turbulent simulations. We tried to point out that the proper term to use here is residual, which indicates the error between the observed points ($m^{DLC}$) and the estimated loads ($m^{Est}$) that are found via linear regression in (14). Normalizing by the mean of the load across turbines provides a qualitative comparison (Fig. 6, bottom, right) between different turbine parts. It's not a perfect metric, as small mean values can be inflated (like Tower Clearance, which was removed from this plot). However, the appropriate values for the residual uncertainty are placed in the figures of Sections 8 to 11. |
| | **Changes in manuscript:** A more detailed explanation and example for computing the turbulent load contribution is provided in Section 6: |
| | [We quantify the turbulent load contribution using the turbulence factor $f^{turb}$] to compare between different turbine parts on how much of the design load $m^{DLC}$ is attributed to turbulent versus harmonic loading. For example, all peak main bearing loads found using DLC simulations are shown in Fig. 6 (top). The average design load ($m^{DLC}$) of the 3-bladed peak main bearing loads (magenta) in Fig. 6 (top, left) is approximately 40 MNm, while the average of the corresponding harmonic loads ($m^H$) is approximately 10 MNm. Thus, the average turbulent load ($m^{turb}$) is |

| | |
|---|---|
| | approximately 30 MNm by (11). Thus, using (12), $f^{turb} \approx 0.75$, as shown in Fig. 6 (left, bottom) along with a selection of the  other turbine loads.

Throughout the article, we have replaced error with residual to better represent its meaning.

The sentence describing the standard deviation of the residual being normalized by the mean has been re-worded to more clearly describe its use:

In Fig. 6 (bottom, right), we normalize the standard deviation of the residual by the mean load over all rotors to compare the fit of the transformation across different turbine parts. |
| 5. In equation 13, does this mean you need two calibration constants for each of the 3 azimuthal modes you are considering? | **Answer:** We use different calibration (renamed as transformation) constants $(a^{trans}, b^{trans})$ that are determined separately for 2- and 3- bladed rotors, each load axis, and both peak and fatigue loads. |
| | **Changes in manuscript:** A sentence was added after equation (13) to clarify this point:

Because 2- and 3-bladed rotors sample turbulence differently, we define a  transformation set ($a^{trans}$; $b^{trans}$) separately for each, illustrated by the different fits of Fig. 6 (top, left). There are also different transformation sets for each design load: at each axis and for both peak and fatigue loads. To estimate the design load, the  transformation set corresponding to the desired component, axis, and number of blades is used: … |
| Figure 9 seems to be showing a lot of interesting trends. It might be useful to inform the reader which design load cases were the driving cases. For example, increasing damage equivalent load but decreasing | **Answer:** Thank you for this comment. Our design goals can be made more clearly. The design driving load for the SUMR-13A was found to be the peak flapwise bending moment. To account for this and to increase power capture, the design goal for the SUMR-13B is to constrain peak flapwise bending moments and increase AEP. Due to its more massive blades, the design driving loads for the SUMR-13B are the edgewise fatigue loads; this leads to the design study in Section 8.2.1. |

| maximum peak load might be ok if tip deflection is the driving DLC. | **Changes in manuscript:** A paragraph was added to Section 8.1 describing the design driving loads of both rotors and the goal for the SUMR-13B: |
|---|---|
| | The SUMR-13A blade design was found to be driven by extreme loading along a combined flapwise and edgewise direction, where DLC 1.4 (extreme coherent gust with direction change) caused the greatest blade load. Since edgewise loads are deterministic, varying with a near constant amplitude with respect to the rotor azimuth, the design goal of the next rotor iteration, the SUMR-13B was to constrain peak flapwise loads and increase power capture using the aerodynamic design changes previously described. The SUMR-13B is not necessarily cost optimal. Using larger blades with both greater power capture and structural loading could result in a net cost benefit compared to the SUMR-13B. However, in the absence of a detailed cost model, these design choices are difficult to make and depend on a wide array of factors. Larger rotors with both increased loading and power capture will be investigated in future design iterations. |
| | The SUMR-13B does, however, provide a demonstration for using the harmonic loads and results in Fig. 9 to guide design: the aerodynamic design changes can be applied in combination. Since the goal of the SUMR-13B is to constrain peak flapwise loads and increase power capture (AEP), some combination of increasing the blade length, decreasing the axial induction, and increasing the cone angle should provide a blade with the desired properties. Looking at the peak flapwise blade load (leftmost in Fig. 9), if we start at the SUMR-13A, the black dot at (1,1), and increase the available rotor power to 16.9 MW, we will have a rotor with the relative power and load at the blue diamond. Then, if we decrease the axial induction to 0.2, the change in power and load is as if only the axial induction (and corresponding blade length increase) were changed by that amount (red, dashed vector). Finally, by increasing the cone angle from 5 deg. to 12.5 deg., the change in power and load is equivalent to the change indicated by the yellow, dashed vector. The combination of these design changes results in the AEP and structural loading of the SUMR-13B: it increases AEP by 11 % compared to the SUMR-13A, while constraining peak blade flapwise loads to the level of the SUMR-13A. The same changes can be applied in combination to the flapwise DELs and edgewise DELs. The increased blade length of the SUMR-13B increases the |

| | flapwise DELs due to the enhanced effect of wind shear and edgewise DELs due to the additional blade weight. During the SUMR-13B structural layup design, we found the design driving blade load to be the fatigue DEL in the edgewise direction, which will be the focus of Sect. 8.2.1. |

**Response to Referee #2**

**MS Number:** wes-2018-80
**Title:** System-level design studies for large rotors
**Corresponding author:** Daniel Zalkind

We would like to thank referee #2 for their review and comments on our research paper. Through a major revision of the manuscript (including a formal internal review by the National Renewable Energy Laboratory), we have tried to address all the referee's comments and we feel the result is a much stronger contribution. The following table collects the referee's comments, the authors' responses to each point, and the authors' changes in the manuscript. In addition, a color-coded version of the manuscript is provided, in which all changes can be easily identified. We have used the red color to indicate text that has been removed from the submitted manuscript. The descriptions in blue represent the added or re-written parts, addressing the referee's comments.

| Comments of Referee #1 | Authors' Responses |
|---|---|
| (1) In section 3, it is unclear what is the justification of assuming that the tower fairing is present in the current simulations. Can the authors be elaborated? | **Answer:** Initially, it was felt that a tower fairing could be important to reduce fatigue loads. However, more detailed recent studies using a validated FAST approach (Noyes *et al*. 2018) have shown the shadow effect for this downwind coned geometry was minimal once turbulent inflow conditions were considered (e.g. changed the DEL values by only a few %).

 To maintain consistency between the harmonic and turbulent load simulations, we selected the simplifying assumption of no tower shadow for both sets of simulations in this study. Therefore, we have neglected the tower shadow effect for all these simulations in order to focus on the influence of the more important harmonic and turbulent loads. |
| | **Changes in manuscript:** We have eliminated the mention of a tower fairing in Section 3:

 A recent FAST-based, wind tunnel validated approach has shown that, compared with turbulence, tower shadow effects are relatively small (Noyes et al., 2018).. Thus, for simplicity, we have  omitted the tower shadow model from our analysis in order to focus on the influence of the more important harmonic and turbulent loads. |

| (2) In section 4, do the authors have an explanation why at different wind speeds, the rotors have different transitions? How [does] that affect the parameters of the control architecture? | **Answer:** The larger SUMR-13B rotor produces greater aerodynamic torque than the SUMR-13A, leading to higher generator speeds at the same below-rated wind speed. Thus, the SUMR-13B will reach rated generator speed and transition to above-rated control at a lower wind speed.

The different transition, or rated, wind speeds do not affect the parameters of the control architecture. All turbines in this article use a gain-scheduled, proportional-integral control system for above-rated pitch control and all have a rated generator speed of 1173.7 rpm. The gearbox ratio of each machine is changed to reflect the different rated rotor speeds in Table 1. Once the generator speed exceeds the rated generator speed, pitch control is activated; this happens at a lower wind speed for larger rotors. |
|---|---|
| | **Changes in manuscript:** Sentences were added in Section 4 to clarify that the same generator model is used for all turbines, but with different gearbox ratios:

The generator rated power of 13.2 MW and rated speed of 1173.7 rpm are assumed to be constant for all the turbines in this study. The gearbox ratio of each turbine is changed to reflect the aerodynamically-optimal rated rotor speed. |
| (3) In section 5, why [do] the authors select only the first four harmonics to reconstruct the load signal? Is that because of the linearity of these harmonics or because of their energy? How would the results change if we consider more or random harmonics? | **Answer:** We analyze only the first four harmonics of the load signal because those harmonics contain most of the information in the signal. Of the first 10 harmonics of a blade load signal in a constant, sheared inflow (harmonic load), the amplitude of the first 4 harmonics accounts for 99.9% of the energy.

Most of the amplitude of wind turbine loads for the harmonic loads is due to the mean load ($m^0$) and the dominant harmonic component: the 1P or $N_B$P load, depending on whether the turbine part is rotating or non-rotating, respectively, where $N_B$ is the number of blades. The magnitude of these harmonics is in the range of ($10^1$–$10^2$ MNm), whereas the non-dominant harmonic loads have a magnitude of about $10^0$ and the magnitudes of the higher harmonic amplitudes are even less.

To reconstruct the load signal and create a mapping to design loads simulated in turbulence, we would like to use as few signals as possible to represent the mapping. More or random harmonics would only change the mapping slightly (due to the small load amplitudes at those harmonics) and make the definitions for peak and fatigue DELs more complicated. |

| | **Changes in manuscript:** A more thorough motivation for deriving harmonic loads is provided in the introduction of Section 5: |
|---|---|
| | In this section, we describe harmonic loads $m^H$, which are derived from constant and periodic loads that arise due to steady wind loading, wind shear, and turbine self-weight. These harmonic loads can be mapped, or transformed, into estimates $m^{Est}$ of design loads $m^{DLC}$ that are computed using operational DLC simulations in Sect. 6. The key simplification of the harmonic load model compared to design loads computed using DLC simulations is the omission of load variations that do not correspond to the rotor speed. These non-harmonic load variations arise because of wind speed and direction changes, as well as the component's natural frequencies. All frequency components are necessary to determine the design load for a final, detailed design, but for exploring potentially large numbers of design trade-offs, simplified harmonic loads provide enough information about the various turbine loads. |
| | A discussion of the load magnitude at each harmonic is also provided in Section 5: |
| | An example for the blade flapwise load is shown in Fig. 4; most of the load magnitude is in the constant $m^0$ and once-per-revolution $m^{1P}$ load component ($10^1$–$10^2$ MNm), with some in the 2P load component due to shaft tilt and gravity (~$10^0$ MNm), and very little in the higher harmonics ($<10^{-1}$ MNm). |
| (4)   In subsection 5.1, the author used the Weibull distribution, which is one of the most widely used lifetime distributions in reliability engineering, can the authors explain what is the range of Wöhler exponent used in the steady state and with turbulence cases? Can the authors explain the effect of the gust or rare-events on the Wöhler exponents and the dominant harmonic components? | **Answer:** The Wöhler exponents used in this study reflect the values commonly used for composite materials (n = 10, blades) and for steel (n = 3, all other components). The same values were used for both the harmonic load (eq. 10) and in the turbulent fatigue calculations using MLife.

Larger Wöhler exponents (e.g., for the blades) place a larger weight on gust or rare-events compared to materials with lower Wöhler exponents. Thus, the blade fatigue is driven more by gust or rare-events than the other components; this was not found to have a large effect on the turbulent load component or residual of the mapping in Section 6. |
| | **Changes in manuscript:** The Wöhler exponents for each component are now listed in Table 2. |

| | |
|---|---|
| (5) In Section 6, it is unclear why the peak loads are more deterministic than DELs and rotating components are more deterministic than non-rotating component loads. | **Answer:** Thank you for this comment. When we say deterministic, we mean that the load has a smaller component due to turbulence and a larger component due to steady and periodic (harmonic) loading. |
| | Peak loads are more well defined by the harmonic loads than DELs because the peak loads are defined by a large steady state component, which is the same for both the turbulent and harmonic loads. The turbulent DELs are affected by dynamic changes in wind speed, which is not modelled by the harmonic loads. |
| | **Changes in manuscript:** Sentences were added and descriptions like "more deterministic" were removed for precision after this statement in Section 6 to provide our explanation of this observation: |
| | Some loads, like the edgewise (Blade X) DEL and the hub DEL about the $z_h$-axis for 2-bladed rotors, are  better represented by the harmonic model, as indicated by lower turbulence factors compared with the others. In general, peak loads are  better represented by the harmonic load than DELs and rotating component loads are  better represented by the harmonic model than non-rotating component loads. Peak loads, defined both by the harmonic model and in turbulent simulations, depend to a large extent on the constant or mean wind speed, respectively, which is represented with the same value in both cases. On the other hand, wind speed changes have a large effect on the fatigue DELs, which is not modeled by the harmonic load. Rotating component loads in turbulence are primarily driven by the 1P load, which is more clearly modeled by the harmonic loads, due to gravity and wind shear, than the smaller $N_B$P load component. |
| (6) In section 6, the authors mentioned that "[some] turbine parts have large turbulent components that are not directly modeled in steady state;" what [do] the authors mean by not directly modeled, and how much [does] this point affect the correlation and standard deviation shown in Fig. 6? | **Answer:** In this statement, when we refer to the steady state loads, we mean that the non-rotating components of 3-bladed rotors have very small 3P loads. Thus, the corresponding harmonic DELs are small, but the turbulent DELs can still be large, which leads to large turbulent factors $f^{turb}$ and increases the standard deviation of the residual in Fig. 6. However, there is still a correlation and the residuals still appear to be acceptable given their unnormalized values (e.g., in Fig. 15). |
| | **Changes in manuscript:** Throughout the article, the use of steady state has been eliminated and replaced by more precise language describing the type of load we are referring to. |

| | |
|---|---|
| (7) In section 6, do the linear calibration factors relate to selecting only the first four harmonics? | **Answer:** Only 1 or 2 harmonics are used to determine $m^H$. Given the relative amplitude of the harmonics not included in the computation of $m^H$, the effect on the calibration factors would be very small. |
| | **Changes in manuscript**: We clarified the motivation for transforming the harmonic loads into design loads in the introduction of Section 5 and provided a discussion of the relative harmonic load magnitudes.

For more information about the associated manuscript updates, please see our response to Referee comment/question #3 above. |
| (8) In section 6, what is the correlation between the uncertainty and level of turbulence? | **Answer:** Thank you for this question. Only Class 2B turbulence was considered in this study. Other studies (Dimitrov, 2018 and Robertson, 2018) show that design loads are sensitive to turbulence intensity. If the turbulence level increased, we would expect the turbulence factor $f^{turb}$ in (11) to increase. Since we have seen a positive correlation between the turbulence factor and uncertainty in the mapping, we would expect the residual to increase with increasing levels of turbulence. |
| | **Changes in manuscript:** In Section 6, a note was added to clarify that Class IIB turbulence is used in the analysis:

We quantify the turbulent load contribution $m^{\text{turb}}$ of each component load using the turbulence factor $f^{turb} = \ldots$ to compare between different turbine parts on how much of the design load $m^{\text{DLC}}$ is attributed to turbulent versus harmonic loading for Class IIB turbulence.

In Section 12 (Model limitations, suggested improvements, and potential use), a sentence was added to mention that this work only considered a single turbulence level and that others would change the turbulence factor and uncertainty of the transformation:

Levels of turbulence, besides Class IIB that was analyzed in this study, would result in a different turbulent component and residual of the transformation from harmonic to design load. |
| (9) In section 6, the authors mentioned that "the most erroneous value (using the metric of standard deviation | **Answer:** The result of the "most erroneous" value being the minimum tower clearance is due to the imperfect metric (the standard deviation normalized by the mean) used to compare the residual error of the transformation from harmonic to design loads for different turbine parts. For the power and |

| normalized by the mean) is the minimum tower clearance, but this is influenced by the small average tower clearance over all rotors." How much is this point affected by the result assuming that the tower fairing is present? | loads compared in Fig. 6 (bottom, right), the mean values that normalize the residual error are all large positive values. |
|---|---|
| | However, for the tower clearance, the values are near-zero and, in some cases, negative (Fig. 14). When normalized by this near-zero value, it seems to imply a poor fit. The unnormalized residual error of the estimated tower clearance is presented in Fig. 14: it has a standard deviation of 1.36 m. Compared to the magnitude of the tower clearances found in in the study represented in Fig. 14, this is an acceptable estimate to gain an understanding for how cone angle affects tower clearance. The study is not intended to replace a detailed DLC analysis that would determine the chance of a tower strike. |
| | In Fig. 14, the tower clearance is the minimum perpendicular distance between the blade tip and the yaw axis. While the tower fairing is no longer needed for this turbine, more detailed simulations are recommended and a detailed tower geometry would be required to deem the tower safe from blade strike. |

**Changes in manuscript:** To reduce confusion and since the metric does not apply well to values with a near-zero mean, we have eliminated tower clearance from the discussion in Section 6 and the bottom, right plot of Fig. 6:

To clarify the precise definition of tower clearance in this study and highlight the extra information needed to determine blade clearance, we have included the following in Sect. 10:

These hub-to-tower overhang values result in adequate tower clearance (the minimum perpendicular distance between the blade tip and the yaw axis $y_z$) when the cone angle is at least 5°  away from the tower (Fig. 14, left). However, such an important design parameter would certainly be subject to verification using a detailed tower design and the full set of DLCs before deeming the tower safe from blade strike.

| (10) In subsection 8.1, can the authors elaborate on the impact of constraining the transition on the above-rated control at lower wind speeds? | **Answer:** The transition to above rated control is not constrained at lower wind speeds. The same control is used for all turbines in this study. When the generator exceeds the rated generator speed (1173.7 rpm for all rotors), the pitch controller causes the pitch angle to increase. This pitch increase reduces loads.

The "constraint" we were referring to is that we would have expected larger rotors to have larger increases in peak blade flapwise loading. By transitioning to above rated control at a lower wind speed than a smaller rotor, this increase in peak blade loads is limited. |
| | **Changes in manuscript:** The control scheme used for all rotors is clarified in Section 4:

We have chosen this  control architecture, which is the same for all rotors, so that it can be easily tuned for  many rotors in the same way.

Additionally, the wording referred to in this question is altered in Section 8:

However, all rotors are controlled to have the same rated generator power of 13.2 MW, which  limits the increase in peak blade loads by transitioning to above-rated control at lower wind speeds. |
| (11) In subsection 8.1, I see a potential innovation in the analysis and comparison should be provided in Fig. 9. | **Answer:** A key potential innovation that we see in this analysis is that the design changes (due to blade length, axial induction, and cone angle) can be applied in combination, which is reflected in Fig. 9. |
| | **Changes in manuscript:** To clarify this innovation, a more thorough description is provided in Subsection 8.1, along with an example for the Blade flapwise peak load:

The SUMR-13B does, however, provide a demonstration for using the harmonic loads and results in Fig. 9 to guide design: the aerodynamic design changes can be applied in combination. Since the goal of the SUMR-13B is to constrain peak flapwise loads and increase power capture (AEP), some combination of increasing the blade length, decreasing the axial induction, and increasing the cone angle should provide a blade with the desired properties. Looking at the peak flapwise blade load (leftmost in Fig. 9), if we start at the SUMR-13A, the black dot at (1,1), and increase the available rotor power to 16.9 MW, we |

will have a rotor with the relative power and load at the blue diamond. Then, if we decrease the axial induction to 0.2, the change in power and load is as if only the axial induction (and corresponding blade length increase) were changed by that amount (red, dashed vector). Finally, by increasing the cone angle from 5 deg. to 12.5 deg., the change in power and load is equivalent to the change indicated by the yellow, dashed vector. The combination of these design changes  result in the AEP and structural loading of the SUMR-13B: it increases AEP by 11 % compared to the SUMR-13A, while constraining peak blade flapwise loads to the level of the SUMR-13A. The same changes can be applied in combination to the flapwise DELs and edgewise DELs. The increased blade length of the SUMR-13B increases the flapwise DELs due to the enhanced effect of wind shear and edgewise DELs due to the additional blade weight. During the SUMR-13B structural layup design, we found the design driving blade load to be the fatigue DEL in the edgewise direction, which will be the focus of Sect. 8.2.1.

| (12) In section 8.1, the authors highlighted that the blades of three-bladed rotors experience lower loads in comparison with two-bladed rotors, it is unclear if this is the case for the damage equivalent load; can the authors explain that? | **Answer:** Fig. 9 shows that the blade loads for the 3-bladed rotors are less than the loads for similar 2-bladed rotors for blade flapwise peak, fatigue, and edgewise fatigue loads. This is primarily due to the larger chord length and mass of the blades in a 2-bladed rotor versus a similarly powered 3-bladed rotor. |
| --- | --- |
| | **Changes in manuscript:** A note has been added in Section 8.1 to make this result clear:

The blades of the three-bladed rotors experience lower loads (both peak and fatigue, edgewise and flapwise) with the same power capture due to their smaller chord and mass.

In the caption of Fig. 9, we added a sentence referencing the 3-bladed rotor designs:

The set of 3-bladed rotor designs are represented with a dotted curve. |

| | |
|---|---|
| (13) In subsection 9.1, please can the authors explain how 2P harmonic loads are used to determine the 3P load? Also, what is the impact of the 2P and 3P on rotating and non-rotating parts? Did this analysis include the effect of steady-state and turbulence? Please elaborate. | **Answer:** 2P harmonic loads map to 3P loads by $$m_{sy}^{3P} = \frac{1}{2}\left(m_{hy,c}^{2P} + m_{hz,s}^{2P}\right)$$ using the rotation matrix in eq. (23). The 2P loads affect the non-rotating components of 2-bladed turbines and the 3P loads affect the non-rotating components of 3-bladed rotors. The effect the 2P and 3P harmonics on the rotating components is relatively small for both harmonic (steady-state) and turbulence loads compared to 1P loads. |
| | **Changes in manuscript:** A sentence has been expanded in Section 9 for the derivation of 3P loads: The 3P component is determined similarly based on the 2P harmonic load components by using (23). A comparison of the load harmonics at the different periodic frequencies is provided in Section 5: for more information about this comparison and associated manuscript updates, please see our response to Referee comment/question #3 above. |
| (14) In section 9.2, it is unclear why the authors omitted the turbulent component from the calibration set of 2-bladed rotors. | **Answer:** The reason we omitted the free teeter hinge from the transformation set is because it behaves much differently than the other rotors in terms of the harmonic and turbulent contributions to the design load. Turbulence has a much smaller effect, compared with all the other rotors. In Fig. 6, it is shown as an outlier. |
| | **Changes in manuscript:** A sentence has been added to Section 9.2 to clarify this decision: In other words, the design loads for the main bearing are nearly equal to the  harmonic loads, but in every other case there is a significant turbulent component, as mentioned in Sect. 6.  Since this case is an outlier and behaves differently when mapping harmonic loads to turbulent loads, it is omitted from the  transformation set of 2-bladed rotors. |

| | |
|---|---|
| (15) In subsection 9.3, can the authors highlight the turbulence contribution on the peak load on the main bearing and the other non-rotating components in terms of the cone angle and combine that with the yaw bearing and tower? | **Answer:** The turbulence contribution is highlighted by providing the standard deviation of the residual of the mapping from harmonic to design load, where it was applied, in each plot of Figs. 13—15. The authors believe there is too much information presented in each plot to adequately combine them. No general conclusions can be made on the effect of the cone angle on the turbulence factor of the non-rotating components. |
| (16) Can the authors include the citations for the equations used in this study? | **Answer:** Most equations have been developed by the authors for the study that this paper documents. Where the equations are not originally derived, defined, or based on common knowledge, citations have been added. |
| | **Changes in manuscript:** Citations have been added for eq. (2), (3–6), and (20). |

**System-level design studies for large rotors**

Daniel S. Zalkind[1], Gavin K. Ananda[2], Mayank Chetan[3], Dana P. Martin[4], Christopher J. Bay[5], Kathryn E. Johnson[4,5], Eric Loth[6], D. Todd Griffith[3], Michael S. Selig[2], and Lucy Y. Pao[1]

[1]Department of Electrical, Computer & Energy Engineering, University of Colorado Boulder, Boulder, CO 80309, USA
[2]Department of Aerospace Engineering, University of Illinois Urbana-Champaign, Champaign, IL 61820, USA
[3]Department of Mechanical Engineering, University of Texas at Dallas, Richardson, TX 75080, USA
[4]Department of Electrical Engineering, Colorado School of Mines, Golden, CO 80401, USA
[5]National Wind Technology Center, National Renewable Energy Laboratory, Golden, CO 80401, USA
[6]Department of Mechanical and Aerospace Engineering, University of Virginia, Charlottesville, VA 22904, USA

**Correspondence:** Daniel S. Zalkind (dan.zalkind@gmail.com)

**Abstract.** We examine the effect of rotor design choices on the power capture and structural loading of each major wind turbine component. A  harmonic model for structural loading is derived from simulations using the NREL aeroelastic code FAST  to reduce computational expense while evaluating design trade-offs for rotors with radii greater than 100 m. Design studies are performed, which focus on blade aerodynamic and structural parameters as well as different hub configurations and nacelle placements atop the tower. The effects of tower design and closed-loop control are also analyzed. Design loads are calculated according to the IEC design standards and used to  create a mapping from the harmonic model  of the loads and quantify the uncertainty of the transformation.

Our design studies highlight both industry trends and innovative designs: we progress from a conventional, upwind, 3-bladed rotor, to a rotor with longer, more slender blades that is downwind and 2-bladed. For a 13 MW design, we show that increasing the blade length by 25 m while decreasing the induction factor of the rotor increases annual energy capture by 11 % while constraining peak blade loads. A downwind, 2-bladed rotor design is analyzed, with a focus on its ability to reduce peak blade loads by 10 % per 5 deg. of cone angle, and also reduce total blade mass. However, when compared to conventional, 3-bladed, upwind designs, the peak main bearing load of the up-scaled, downwind, 2-bladed rotor is increased by 280 %. Optimized teeter configurations and individual pitch control can reduce non-rotating damage equivalent loads by 45 % and 22 %, respectively, compared with fixed-hub designs.

*Copyright statement.* Christopher J. Bay's copyright for this publication is transferred to Alliance for Sustainable Energy, LLC.

[revised manuscript text omitted]